# Impact of seeder-feeder cloud interaction on precipitation formation: a case study based on extensive remote-sensing, in-situ and model data

Kevin Ohneiser<sup>1</sup>, Patric Seifert<sup>1</sup>, Willi Schimmel<sup>1</sup>, Fabian Senf<sup>1</sup>, Tom Gaudek<sup>1</sup>, Martin Radenz<sup>1</sup>, Audrey Teisseire<sup>1</sup>, Veronika Ettrichrätz<sup>2</sup>, Teresa Vogl<sup>2</sup>, Nina Maherndl<sup>3</sup>, Nils Pfeifer<sup>2</sup>, Jan Henneberger<sup>4</sup>, Anna J. Miller<sup>4</sup>, Nadja Omanovic<sup>4</sup>, Christopher Fuchs<sup>4</sup>, Huiying Zhang<sup>4</sup>, Fabiola Ramelli<sup>4</sup>, Robert Spirig<sup>4</sup>, Anton Kötsche<sup>2</sup>, Heike Kalesse-Los<sup>2</sup>, Maximilian Maahn<sup>2</sup>, Heather Corden<sup>5</sup>, Alexis Berne<sup>5</sup>, Majid Hajipour<sup>1</sup>, Hannes Griesche<sup>1</sup>, Julian Hofer<sup>1</sup>, Ronny Engelmann<sup>1</sup>, Annett Skupin<sup>1</sup>, Albert Ansmann<sup>1</sup>, and Holger Baars<sup>1</sup>

**Correspondence:** K. Ohneiser (ohneiser@tropos.de)

#### Abstract.

A comprehensive approach to study the seeder-feeder mechanism in unprecedented detail from a combined remote-sensing, in-situ, and model perspective is shown. This publication aims at investigating the role of the interplay of a seeder-feeder cloud system and its influence on precipitation formation based on a case study from 8 Jan 2024 observed over the Swiss Plateau in Switzerland.

This case study offers an ideal setup for applying several advanced remote-sensing techniques and retrieval algorithms, including fall streak tracking, radar Doppler peak separation, dual-wavelength radar applications, a liquid detection retrieval, a riming retrieval, and an ice crystals shape retrieval. Results indicate that a large portion of ice mass was rimed, which is attributed to persistent coexistence of falling ice crystals and supercooled water within low-level supercooled liquid water layers. Interaction of seeder and feeder clouds results in a significant precipitation enhancement. This has implications on the water cycle. From the anti-correlation between surface precipitation and liquid water path we estimated that 20–40% of the precipitation stems from the feeder cloud. However, we have to note that the value of 20–40% is strongly dependent on the assumed reproduction rate of liquid water in the feeder cloud. It is also found that in this specific case study precipitation was significantly underestimated by the operational ICON-D2 model runs during the seeder-feeder process. Contrarily, during periods when the cloud system does not interact, precipitation is significantly overestimated by the model in the case study.

This study aims at giving an overview from a remote-sensing, in-situ and model perspective on a seeder-feeder event in an unprecedented detail by exploiting a big set of retrievals applicable to remote-sensing and in situ data. Utilizing different retrievals gives a consistent view on the seeder-feeder case study which is an important basis for future studies. It is demonstrated

<sup>&</sup>lt;sup>1</sup>Leibniz Institute for Tropospheric Research (TROPOS), Leipzig, Germany

<sup>&</sup>lt;sup>2</sup>Leipzig Institute for Meteorology (LIM), Leipzig University, Leipzig, Germany

<sup>&</sup>lt;sup>3</sup>Faculty of Civil Engineering and Geosciences, TU Delft, Delft, Netherlands

<sup>&</sup>lt;sup>4</sup>Institute for Atmospheric and Climate Science, ETH Zurich, Zurich, Switzerland

<sup>&</sup>lt;sup>5</sup>Environmental Remote Sensing Laboratory, EPFL, Lausanne, Switzerland

how improved understanding of seeder-feeder interactions can contribute to enhancing weather forecast models, particularly in regions affected by persistent low-level supercooled stratus clouds.

# 1 Introduction

Precipitation formation in mid-latitudes is predominantly driven by mixed-phase clouds processes, where ice and liquid water coexist. Studies show that 60–90% of precipitation in these regions originates from such clouds, making them the dominant pathway for precipitation generation in mid-latitudes (Lau and Wu, 2003, 2011; Mülmenstädt et al., 2015; Korolev et al., 2017). One important mechanism within these clouds is the seeder-feeder mechanism, known to significantly enhance precipitation and thus play a critical role in the Earth's water cycle (Purdy et al., 2005; Heymsfield et al., 2020).

Seeder clouds, which can be pure ice or mixed-phase clouds themselves, produce ice crystals, for example supported by ice-nucleating particles (INP), that fall into feeder clouds below (Ramelli et al., 2021a). Feeder clouds, acting as a moisture reservoir, typically mainly consist of supercooled liquid cloud droplets that contribute to the growth of falling ice crystals or to an enhancement of particle number and ice mass.

There are several processes that can lead to an enhanced ice mass or ice crystal number concentration (ICNC). The aggregation process is for example most efficient at temperatures around –14°C (dendritic growth) and close to 0°C (sintering) (Hosler et al., 1957). Riming occurs when ice crystals fall through a layer of supercooled liquid water (Erfani and Mitchell, 2017).

The supercooled water freezes immediately on available ice crystals and makes them heavier and more spherical in shape. The Hallett-Mossop process, also called rime splintering, is most efficient at temperatures between –3 and –8°C and enhances the number of ice crystals (Hallett and Mossop, 1974; Mossop and Hallett, 1974). This process is responsible for secondary ice crystal formation pathways (SIP). The Wegener-Bergeron-Findeisen (WBF) WBF process (Wegener, 1911; Bergeron, 1935; Findeisen, 1938), where water vapor preferentially deposits onto ice crystals at the expense of supercooled droplets, accelerates ice growth and enhances precipitation.

The interaction of seeder-feeder cloud systems also has an influence on cloud lifetime and cloud radiative effects. Supercooled liquid droplets in mixed-phase clouds are more opaque to longwave radiation and increases cloud albedo more than ice crystals which has consequences on radiative properties of cloud systems (Hogan et al., 2003). Matus and L'Ecuyer (2017) describe that liquid clouds lead to a negative global radiative contribution of  $-11.8 \, \mathrm{W \, m^{-2}}$ , ice clouds have a positive radiative effect of  $3.5 \, \mathrm{W \, m^{-2}}$ , and multilayered clouds with distinct layers of liquid and ice exert a negative radiative effect of  $-5.4 \, \mathrm{W \, m^{-2}}$ . Their conclusion is that it is essential to accurately represent mixed-phase clouds in future climate scenarios for quantifying cloud feedbacks.

While the seeder-feeder mechanism can enhance precipitation by 20–50% in some regions (Ramelli et al., 2021a), it remains difficult to accurately simulate it in weather forecast models. Models often struggle to capture the exact balance between ice and liquid water in mixed-phase clouds, which leads to significant errors in precipitation forecasts (Klein et al., 2009; Schemann and Ebell, 2020; Kiszler et al., 2024). Specifically, models tend to overestimate ice formation, which reduces longevity

of mixed-phase clouds and leads to precipitation being underestimated during seeder-feeder interactions. Detailed observational studies in combination with high-resolution model simulations help to shed light on factors influencing cloud-phase partitioning. Kalesse et al. (2016a) studied a low-level mixed-phase stratiform cloud case observed over Barrow, Alaska. They find major influences on the cloud system caused by large-scale advection of different air masses with different aerosol concentrations and humidity content, cloud-scale processes such as a change in thermodynamical coupling state, and local-scale dynamics influencing the residence time of ice crystals.

Recent studies focus on seeder-feeder events, primarily from the perspective of remote-sensing and model simulations (Robichaud and Austin, 1988; Purdy et al., 2005; Arulraj and Barros, 2019; Vassel et al., 2019; Ramelli et al., 2021a; Proske et al., 2021; Misumi et al., 2021; He et al., 2022; Dedekind et al., 2023; Di and Yuan, 2023). However, a detailed analysis that combines remote-sensing, in-situ measurements, and ICON-D2 model (Icosahedral Nonhydrostatic Modell Zängl et al., 2015; Omanovic et al., 2024) data to study natural seeder-feeder events remains limited. This study aims to address this gap by analyzing a natural seeder-feeder event in unprecedented detail, using a very large synergistic multi-frequency radar, lidar, and in-situ observation campaign in Europe.

Findings from this work have the potential to improve the representation of seeder-feeder processes in weather prediction models, particularly in regions where persistent low-level stratus clouds frequently occur.

The CLOUDLAB (Henneberger et al., 2023) campaign and PolarCAP (Polarimetric Radar Signatures of Ice Formation Pathways from Controlled Aerosol Perturbations) project, in the frame of PROM (Polarimetric Radar Observations meet Atmospheric Modelling PROM, 2024) were conducted in Eriswil, Switzerland, during the winters of 2022/23 and 2023/24. A comprehensive dataset is provided that allows for a detailed investigation of the seeder-feeder interaction. By integrating remote-sensing data, in-situ measurements, and numerical models, this study offers new insights into the microphysical processes causing precipitation enhancement. A fall streak tracking algorithm is applied to trace the evolution of microphysical properties along the path of falling ice crystals, providing valuable information on how seeder and feeder clouds interact. The fall streak is based on the maximum of the effective radar reflectivity  $Z_{\rm e}$  at different height levels.

Section 2 presents the experimental setup. Section 3 deals with the applied methods in the study. In Sect. 4 the weather situation on 8 Jan 2024 is described. Section 5 focuses on remote-sensing and in-situ observations. Section 6 focuses on the performance of the ICON-D2 model compared to observations and Sect. 7 will summarize and conclude the results.

# 2 Experimental setup during the winter campaigns in Eriswil

70

In winter seasons 2022/23 and 2023/24, the mobile exploratory platform LACROS (Leipzig Aerosol and Clouds Remote Observations System Radenz et al., 2021) operated by TROPOS (Leibniz Institute for Tropospheric Research) was part of a series of winter campaigns near Eriswil (47.071°N, 7.874°E, 920 m a.s.l) in the Swiss Plateau in the centre of Switzerland. LACROS joined both 3-months campaigns, which were conducted under the umbrella of the ERC (European Research Council) research project CLOUDLAB of ETH Zurich and in the framework of PolarCAP (Polarimetric Radar Signatures of Ice Formation Pathways from Controlled Aerosol Perturbations) project. An overview of the campaign setup can be seen in Fig. 1 and an overview

The CLOUDLAB campaign involved a unique set of ground-based and airborne in-situ cloud and precipitation sensors and

**Figure 1.** a) Topographic map of Switzerland. The red triangle highlights the location of Eriswil. The other triangles correspond to the places used for comparison in the model study in Sect. 6.2 (blue triangle: Huttwil, yellow triangle: Egolzwil, black triangle: Affoltern, grey triangle: Napf). b) Experimental setup of the instruments in Eriswil. Number 18 (Holimo) and 19 (Windsond) are mounted to the HoloBalloon, the windsond can also be launched individually. Number 20 is at Ryseralp, 47.064°N, 7.839°E, approximately 2.7 km away from the main site. All other numbers are explained in more detail in Tbl. 1. Photo: Jan Henneberger.

remote-sensing instruments. During two wintertime campaigns between 2022 and 2024, LACROS enhanced the remote sensing capabilities of the CLOUDLAB campaign with a large number of ground-based equipment, such as a scanning 35-GHz and vertically-pointing 94-GHz cloud radar from TROPOS, Raman polarization lidar, Doppler lidar, ceilometer, micro rain radar, photometer, disdrometer, and microwave radiometer.

During the campaign 2023/24 the instrument site was further enhanced by two additional cooperations. Firstly, the PROM (PROM, 2024) project CORSIPP (Characterization of orography-influenced riming and secondary ice production and their effects on precipitation rates using radar polarimetry and Doppler spectra CORSIPP, 2024) of LIM (Leipzig Institute for Meteorology) joined the campaign in Eriswil with a scanning 94-GHz polarimetric cloud radar (which was placed at Ryseralp, 47.064°N, 7.839°E, approximately 2.7 km southwest of the main site) and the Video In Situ Snowfall Sensor (VISSS Maahn et al., 2024). Secondly, EPFL (École Polytechnique Fédérale de Lausanne) joined the campaign with a scanning polarimetric X-band radar. In combination, the campaign was a very large joint deployments of multi-wavelength radar and lidar systems.

**Table 1.** Description of the measurement instruments shown in Fig. 1. For radars the info vertically pointing (vpt) and scanning mode (scm) is added.

| Ferrone et al., 2022)   Parsivel? Id-disdrometer (Id-differ-Mang and Joss, 2000)   λ = 650 nm                                                                                                                                                                                                                                                                                                                                                                                                                                                                                                                                                                                                                                                                                                                                                                                                                                                                                                                                                                                                                                                                                                                                                                                                                                                                                                                                                                                                                                                                                                                                                                                                                                                                                                                                                                                                                                                                                                                                                                                                                                 | Idx | Instrument (reference)                                                   | Frequency $\nu$ Wavelength $\lambda$ | Quantity                                                                      |
|-------------------------------------------------------------------------------------------------------------------------------------------------------------------------------------------------------------------------------------------------------------------------------------------------------------------------------------------------------------------------------------------------------------------------------------------------------------------------------------------------------------------------------------------------------------------------------------------------------------------------------------------------------------------------------------------------------------------------------------------------------------------------------------------------------------------------------------------------------------------------------------------------------------------------------------------------------------------------------------------------------------------------------------------------------------------------------------------------------------------------------------------------------------------------------------------------------------------------------------------------------------------------------------------------------------------------------------------------------------------------------------------------------------------------------------------------------------------------------------------------------------------------------------------------------------------------------------------------------------------------------------------------------------------------------------------------------------------------------------------------------------------------------------------------------------------------------------------------------------------------------------------------------------------------------------------------------------------------------------------------------------------------------------------------------------------------------------------------------------------------------|-----|--------------------------------------------------------------------------|--------------------------------------|-------------------------------------------------------------------------------|
| A = 050 nm                                                                                                                                                                                                                                                                                                                                                                                                                                                                                                                                                                                                                                                                                                                                                                                                                                                                                                                                                                                                                                                                                                                                                                                                                                                                                                                                                                                                                                                                                                                                                                                                                                                                                                                                                                                                                                                                                                                                                                                                                                                                                                                    | 1   |                                                                          | $\nu = 24  \mathrm{GHz}$             | reflectivity, particle number concentra-<br>tion                              |
| Mira35 STSR cloud radar MBR7 (vpt/scm) (Görsdorf et al., 2015)   ν=51.0-58.0 GHz   ter vapor, brightness temperatures   differential reflectivity, Doppler velocity (vpt/scm) (Görsdorf et al., 2015)   ν=35 GHz   ter vapor, brightness temperatures   differential reflectivity, Doppler velocity (vpt/scm) (Görsdorf et al., 2015)   ν=35 GHz   ter vapor, brightness temperatures   differential reflectivity, Doppler velocity (vpt/scm) (Girsdorf et al., 2015)   ν=35 GHz   ter vapor, brightness temperatures   differential reflectivity, Doppler velocity (vpt/scm) (Görsdorf et al., 2015)   ν=35 GHz   ter vapor, brightness temperatures   differential reflectivity, Doppler velocity, Doppler velocity, Oppler velocity, Oppler velocity, correlation coefficient   v=340-1064 nm   aerosol optical thickness   Doppler velocity, attenuated backscatter cf.   backscatt   | 2   |                                                                          | $\lambda = 650  \mathrm{nm}$         | rain rate, particle number concentration                                      |
| CE318-T Solar lumar photometer (Barreto et al., 2016)   Deppler velocity, attenuated backscatter cf.                                                                                                                                                                                                                                                                                                                                                                                                                                                                                                                                                                                                                                                                                                                                                                                                                                                                                                                                                                                                                                                                                                                                                                                                                                                                                                                                                                                                                                                                                                                                                                                                                                                                                                                                                                                                                                                                                                                                                                                                                          | 3   |                                                                          |                                      | liquid water path (LWP), integrated water vapor, brightness temperatures      |
| Streamline Pro Doppler Lidar (Pearson et al., 2009)  PollyXT Raman polarization lidar (Engelmann et al., 2016)  CHM-15kx Ceilometer (TROPOS) (Wiegner and Geiß, 2012)  Dy two-dimensional video disdrometer (Schönhuber et al., 2008)  RPG94 FMCW-DP cloud radar (vpt) (RPG, 2024)  MASC Multi-Angle Snowflake Camera (Garrett et al., 2012)  Mira35 SLDR cloud radar MBR5 (vpt/scm) (Görsdorf et al., 2015)  Mira35 SLDR cloud radar MBR5 (vpt/scm) (Görsdorf et al., 2015)  CHM-15kx Ceilometer (ETH) (Wiegner and Geiß, 2012)  A=1.64 mm  To poly FMC (All H) (Wiegner and Geiß, 2012)  MASC Multi-Angle Snowflake Camera (Garrett et al., 2015)  Mira35 SLDR cloud radar MBR5 (vpt/scm) (Görsdorf et al., 2015)  Mira35 SLDR cloud radar MBR5 (vpt/scm) (Görsdorf et al., 2015)  Mira35 SLDR cloud radar MBR5 (vpt/scm) (Görsdorf et al., 2015)  Mara4 (HM-15kx Ceilometer (ETH) (Wiegner and Geiß, 2012)  Mara5 (Streamline Pro Dopler velocity, slanted polarization ratio)  Mara6 (HM-15kx Ceilometer (ETH) (Wiegner and Geiß, 2012)  Mara7 (Prosensing, 2024)  HatPRO G5 Microwave radiometer (ETH) (Prosensing, 2024)  Mira8 (Streamline Pro Dopler velocity, differential phase (ETH) (Rose et al., 2005) (Prosensing, 2024)  Mira6 (Prosensing, 2024)  Mira7 (Prosensing, 2024)  Mira8 (Mara7 (Prosensing, 2024)  Mira8 (Mara7 (Prosensing, 2024) (Prosensing, 2024) (Prosensing, 2024) (Prosensing, 2024)  Mira8 (Mara7 (Prosensing, 2024) (Prosensing, 20 | 4   |                                                                          | ν=35 GHz                             | differential reflectivity, Doppler velocity, correlation coefficient          |
| PollyXT Raman polarization lidar (Engelmann et al., 2016)   Er ef.                                                                                                                                                                                                                                                                                                                                                                                                                                                                                                                                                                                                                                                                                                                                                                                                                                                                                                                                                                                                                                                                                                                                                                                                                                                                                                                                                                                                                                                                                                                                                                                                                                                                                                                                                                                                                                                                                                                                                                                                                                                            | 5   |                                                                          | λ=340–1064 nm                        | aerosol optical thickness                                                     |
| CEngelmann et al., 2016   X=353, 532, 1004 nm                                                                                                                                                                                                                                                                                                                                                                                                                                                                                                                                                                                                                                                                                                                                                                                                                                                                                                                                                                                                                                                                                                                                                                                                                                                                                                                                                                                                                                                                                                                                                                                                                                                                                                                                                                                                                                                                                                                                                                                                                                                                                 | 6   |                                                                          | $\lambda$ =1.5 $\mu$ m               | Doppler velocity, attenuated backscatter cf.                                  |
| Wiegner and Geiß, 2012)   X=1064 nm   attenuated backscatter cf.                                                                                                                                                                                                                                                                                                                                                                                                                                                                                                                                                                                                                                                                                                                                                                                                                                                                                                                                                                                                                                                                                                                                                                                                                                                                                                                                                                                                                                                                                                                                                                                                                                                                                                                                                                                                                                                                                                                                                                                                                                                              | 7   |                                                                          | $\lambda$ =355, 532, 1064 nm         | backscatter cf., extinction cf., linear depolarization ratio                  |
| 9drometer (Schönhuber et al., 2008)white lightdrometeor shape, type, size, oblatenes10RPG94 FMCW-DP cloud radar (vpt) (RPG, 2024)ν=94 GHzreflectivity, Doppler velocity, slanted linear depolarization ratio particle number concentration, hy drometeor shape, type, size, oblateness aspect ratio11VISSS Video In Situ Snowfall Sensor (Maahn et al., 2024)λ=530 nmdrometeor shape, type, size, oblateness aspect ratio particle number concentration, hy drometeor shape, type, size, oblateness aspect ratio12MASC Multi-Angle Snowflake Camera (Garrett et al., 2012)ν=35 GHzreflectivity, Doppler velocity, slanted linear depolarization ratio13Mira35 SLDR cloud radar MBR5 (vpt/scm) (Görsdorf et al., 2015)ν=35 GHzreflectivity, Doppler velocity, slanted linear depolarization ratio14CHM-15kx Ceilometer (ETH) (Wiegner and Geiß, 2012)λ=1064 nmattenuated backscatter cf.15Digitel DPA-14 INP sampler-INP concentrations16HATPRO G5 Microwave radiometer (ETH) (Rose et al., 2005)ν = 22.24-31.4 GHz por, brightness temperatures reflectivity, Doppler velocity, differential reflectivity, Correlation coefficient differential phase17Venous et al., 2005)ν = 9.385 GHzliquid water path, integrated water val., por, brightness temperatures reflectivity, correlation coefficient differential phase18Holimo Holographic imager for microscopic objects with the helium-filled balloon "Bob" (Henneberger et al., 2013)λ=532 nmparticle number concentration19Windsond S1H3 (Bessardon et al., 2019)-temperature, pressure, relative humid ity, wind speed, wind direction20 <td< td=""><td>8</td><td></td><td>λ=1064 nm</td><td>attenuated backscatter cf.</td></td<>                                                                                                                                                                                                                                                                                                                                                                                                                                     | 8   |                                                                          | λ=1064 nm                            | attenuated backscatter cf.                                                    |
| VISSS Video In Situ Snowfall Sensor (Maahn et al., 2024)  MASC Multi-Angle Snowflake Camera (Garrett et al., 2012)  Mira35 SLDR cloud radar MBR5 (vpt/scm) (Görsdorf et al., 2015)  CHM-15kx Ceilometer (ETH) (Wiegner and Geiß, 2012)  A=1064 nm  HATPRO G5 Microwave radiometer (ETH) (Rose et al., 2005)  TATION (Prosensing, 2024)  Holimo Holographic imager for microscopic objects with the helium-filled balloon "Bob" (Henneberger et al., 2013)  Windsond S1H3 (Bessardon et al., 2019)  N=94 GHz  linear depolarization ratio particle number concentration, hy drometeor shape, type, size, oblateness aspect ratio reflectivity, Doppler velocity, slanted inear depolarization ratio particle number concentration, hy drometeor shape, type, size, oblateness aspect ratio reflectivity, Doppler velocity, slanted inear depolarization ratio particle number concentration, hy drometeor shape, type, size, oblateness aspect ratio reflectivity, Doppler velocity, slanted inear depolarization ratio particle number concentration, hy drometeor shape, type, size, oblateness aspect ratio reflectivity, Doppler velocity, slanted inear depolarization ratio particle number concentration in particle number c | 9   |                                                                          | white light                          | particle number concentration, hy-<br>drometeor shape, type, size, oblateness |
| tometeor shape, type, size, oblateness aspect ratio particle number concentration, hy drometeor shape, type, size, oblateness aspect ratio particle number concentration, hy drometeor shape, type, size, oblateness aspect ratio particle number concentration, hy drometeor shape, type, size, oblateness aspect ratio  MASC Multi-Angle Snowflake Camera (Garrett et al., 2012)  Mira35 SLDR cloud radar MBR5 (vpt/scm) (Görsdorf et al., 2015)  CHM-15kx Ceilometer (ETH) (Wiegner and Geiß, 2012)  A=1064 nm  TINP concentrations  INP concentrations  INP concentrations  Input character of the properties of the properties of the particle number concentration aspect ratio  attenuated backscatter of.  INP concentrations  Input character of the particle number aspect ratio particle number concentration aspect ratio  attenuated backscatter of.  INP concentrations  Input character of the particle number concentration aspect ratio  attenuated backscatter of.  INP concentrations  Input character of the particle number concentration aspect ratio  attenuated backscatter of.  INP concentrations  Input character of the particle number concentration aspect ratio  attenuated backscatter of.  INP concentrations  Input character of the particle number concentration aspect ratio  attenuated backscatter of.  INP concentrations  Input character of the particle number concentration aspect ratio  attenuated backscatter of.  INP concentrations  Input character of the particle number concentration aspect ratio  attenuated backscatter of.  INP concentrations  Input character of the particle number concentration aspect ratio  attenuated backscatter of.  INP concentrations  Input character of the particle number concentration aspect ratio  attenuated backscatter of.  INP concentration  Input character of the particle number concentration aspect ratio  attenuated backscatter of.  INP concentration  Input character of the particle number concentration aspect ratio  attenuated backscatter of.  Input character of the particle number concentration aspect ra | 10  |                                                                          | <i>ν</i> =94 GHz                     | reflectivity, Doppler velocity, slanted linear depolarization ratio           |
| MASC Multi-Angle Showflake Camera (Garrett et al., 2012)λ=1.64 mmdrometeor shape, type, size, oblateness aspect ratio13Mira35 SLDR cloud radar MBR5 (vpt/scm) (Görsdorf et al., 2015)ν=35 GHzreflectivity, Doppler velocity, slanted linear depolarization ratio14CHM-15kx Ceilometer (ETH) (Wiegner and Geiß, 2012)λ=1064 nmattenuated backscatter cf.15Digitel DPA-14 INP sampler–INP concentrations16HATPRO G5 Microwave radiometer (ETH) (Rose et al., 2005)ν = 22.24–31.4 GHz por, brightness temperatures reflectivity, Doppler velocity, differential reflectivity, Doppler velocity, differential phase17StXPol X-band radar (vpt/scm) (Prosensing, 2024)ν=9.385 GHztial reflectivity, correlation coefficient differential phase18Holimo Holographic imager for microscopic objects with the helium-filled balloon "Bob" (Henneberger et al., 2013)λ=532 nmparticle number concentration19Windsond S1H3 (Bessardon et al., 2019)–temperature, pressure, relative humid ity, wind speed, wind direction20RPG94 FMCW-DP cloud radar (LIM)ν=9.4 GHzreflectivity, Doppler velocity, slanted                                                                                                                                                                                                                                                                                                                                                                                                                                                                                                                                                                                                                                                                                                                                                                                                                                                                                                                                                                                                                              | 11  |                                                                          | λ=530 nm                             | drometeor shape, type, size, oblateness, aspect ratio                         |
| 13                                                                                                                                                                                                                                                                                                                                                                                                                                                                                                                                                                                                                                                                                                                                                                                                                                                                                                                                                                                                                                                                                                                                                                                                                                                                                                                                                                                                                                                                                                                                                                                                                                                                                                                                                                                                                                                                                                                                                                                                                                                                                                                            | 12  |                                                                          | $\lambda$ =1.64 mm                   | drometeor shape, type, size, oblateness,                                      |
| (Wiegner and Geiß, 2012)  15 Digitel DPA-14 INP sampler  16 HATPRO G5 Microwave radiometer (ETH) (Rose et al., 2005)  17 StXPol X-band radar (vpt/scm) (Prosensing, 2024)  Holimo Holographic imager for microscopic objects with the helium-filled balloon "Bob" (Henneberger et al., 2013)  19 Windsond S1H3 (Bessardon et al., 2019)  RPG94 FMCW-DP cloud radar (LIM) $\lambda = 1064 \text{ nm}$ attenuated backscatter cf.  INP concentrations  liquid water path, integrated water value por, brightness temperatures reflectivity, Doppler velocity, differential reflectivity, correlation coefficient differential phase $\lambda = 532 \text{ nm}$ particle number concentration $\lambda = 532 \text{ nm}$ temperature, pressure, relative humidity, wind speed, wind direction reflectivity, Doppler velocity, slanted                                                                                                                                                                                                                                                                                                                                                                                                                                                                                                                                                                                                                                                                                                                                                                                                                                                                                                                                                                                                                                                                                                                                                                                                                                                                                            | 13  |                                                                          | ν=35 GHz                             | reflectivity, Doppler velocity, slanted linear depolarization ratio           |
| HATPRO G5 Microwave radiometer $\nu = 22.24-31.4\text{GHz}$ liquid water path, integrated water varieties $\nu = 51.0-58.0\text{GHz}$ por, brightness temperatures reflectivity, Doppler velocity, differential phase reflectivity, correlation coefficient differential phase particle number concentration  HATPRO G5 Microwave radiometer $\nu = 22.24-31.4\text{GHz}$ liquid water path, integrated water varieties por, brightness temperatures reflectivity, Doppler velocity, differential reflectivity, correlation coefficient differential phase with the helium-filled balloon "Bob" (Henneberger et al., 2013)  Windsond S1H3 (Bessardon et al., 2019)  RPG94 FMCW-DP cloud radar (LIM) $\nu = 94\text{GHz}$ temperature, pressure, relative humid ity, wind speed, wind direction reflectivity, Doppler velocity, slanted                                                                                                                                                                                                                                                                                                                                                                                                                                                                                                                                                                                                                                                                                                                                                                                                                                                                                                                                                                                                                                                                                                                                                                                                                                                                                        | 14  | · · · · · · · · · · · · · · · · · · ·                                    | λ=1064 nm                            | attenuated backscatter cf.                                                    |
| (ETH) (Rose et al., 2005) $\nu = 51.0-58.0\mathrm{GHz}$ por, brightness temperatures reflectivity, Doppler velocity, differential reflectivity, Doppler velocity, differential reflectivity, correlation coefficient differential phase17(Prosensing, 2024) $\nu = 9.385\mathrm{GHz}$ tial reflectivity, correlation coefficient differential phase18Holimo Holographic imager for microscopic objects with the helium-filled balloon "Bob" (Henneberger et al., 2013) $\lambda = 532\mathrm{nm}$ particle number concentration19Windsond S1H3 (Bessardon et al., 2019)temperature, pressure, relative humid ity, wind speed, wind direction20RPG94 FMCW-DP cloud radar (LIM) $\nu = 94\mathrm{GHz}$ reflectivity, Doppler velocity, slanted                                                                                                                                                                                                                                                                                                                                                                                                                                                                                                                                                                                                                                                                                                                                                                                                                                                                                                                                                                                                                                                                                                                                                                                                                                                                                                                                                                                  | 15  | Digitel DPA-14 INP sampler                                               | _                                    | INP concentrations                                                            |
| 17 (Prosensing, 2024)                                                                                                                                                                                                                                                                                                                                                                                                                                                                                                                                                                                                                                                                                                                                                                                                                                                                                                                                                                                                                                                                                                                                                                                                                                                                                                                                                                                                                                                                                                                                                                                                                                                                                                                                                                                                                                                                                                                                                                                                                                                                                                         | 16  |                                                                          |                                      | liquid water path, integrated water va-<br>por, brightness temperatures       |
| Holimo Holographic imager for microscopic objects with the helium-filled balloon "Bob" (Henneberger et al., 2013)  19 Windsond S1H3 (Bessardon et al., 2019)  RPG94 FMCW-DP cloud radar (LIM)  20 RPG94 FMCW-DP cloud radar (LIM)  20 reflectivity, Doppler velocity, slanted                                                                                                                                                                                                                                                                                                                                                                                                                                                                                                                                                                                                                                                                                                                                                                                                                                                                                                                                                                                                                                                                                                                                                                                                                                                                                                                                                                                                                                                                                                                                                                                                                                                                                                                                                                                                                                                 | 17  | StXPol X-band radar (vpt/scm) (Prosensing, 2024)                         |                                      | tial reflectivity, correlation coefficient,                                   |
| 2019) — ity, wind speed, wind direction reflectivity, Doppler velocity, slanted                                                                                                                                                                                                                                                                                                                                                                                                                                                                                                                                                                                                                                                                                                                                                                                                                                                                                                                                                                                                                                                                                                                                                                                                                                                                                                                                                                                                                                                                                                                                                                                                                                                                                                                                                                                                                                                                                                                                                                                                                                               | 18  | scopic objects with the helium-filled balloon "Bob" (Henneberger et al., | $\lambda$ =532 nm                    | ·                                                                             |
| RPG94 FMCW-DP cloud radar (LIM) reflectivity, Doppler velocity, slanted                                                                                                                                                                                                                                                                                                                                                                                                                                                                                                                                                                                                                                                                                                                                                                                                                                                                                                                                                                                                                                                                                                                                                                                                                                                                                                                                                                                                                                                                                                                                                                                                                                                                                                                                                                                                                                                                                                                                                                                                                                                       | 19  | · · · · · · · · · · · · · · · · · · ·                                    | _                                    | temperature, pressure, relative humidity, wind speed, wind direction          |
|                                                                                                                                                                                                                                                                                                                                                                                                                                                                                                                                                                                                                                                                                                                                                                                                                                                                                                                                                                                                                                                                                                                                                                                                                                                                                                                                                                                                                                                                                                                                                                                                                                                                                                                                                                                                                                                                                                                                                                                                                                                                                                                               | 20  |                                                                          | ν=94 GHz                             | reflectivity, Doppler velocity, slanted                                       |

#### 3 Data and methods

- In this section the used remote-sensing, in-situ, and model datasets are described. In addition, the applied approaches of the fall streak tracking algorithm, VOODOO (reVealing supercOOled liquiD beyOnd lidar attenuatiOn), dual-wavelength ratio (DWR), Eddy dissipation rate (EDR), peakTree (Doppler-peak-separation algorithm Radenz et al., 2019), ice crystal shape retrieval (Vertical Distribution of Particle Shape, VDPS), riming retrievals, and ice crystal number concentration (ICNC) retrievals will be explained.
- In addition, model results of HYSPLIT (Hybrid Single-Particle Lagrangian Integrated Trajectory) and ICON-D2 are used. Each of the mentioned retrievals contributes to a better understanding of the microphysical processes within the cloud. All retrievals together give a clear picture on the ice crystal habits and changes in ice crystal properties along their way through the cloud.

A separation between liquid water and different habits of ice crystals can be achieved by the analysis of radar Doppler spectra (Radenz et al., 2019). Processes like ice crystal growth, e.g. aggregation or riming, or the determination of ice crystal shapes is done via remote-sensing retrievals developed in recent years. With peakTree it is possible to detect not only dominating ice crystal type but also coexisting ice crystal habits in the same volume. The fall streak tracking algorithm allows us to tell about the history of changes of microphysical properties of ice crystals within clouds (see Fig. 2).

# 115 3.1 Remote-sensing and in-situ data

Remote-sensing techniques allow the creation of vertical profiles of microphysical properties of hydrometeors within a cloud system, offering detailed insights into cloud processes. In combination with ground-based in-situ measurements, remote-sensing observations can be validated. This is very useful as it adds confidence to the retrievals.

The following data sources are utilized in this study:

RPG94 FMCW-DP cloud radar (Radiometer Physics GmbH 94 GHz frequency modulated continuous wave radar - dual polarization) is used for cloud radar reflectivity, spectral width, SLDR, Doppler spectra, and dual-wavelength ratio measurements. The RPG94 minimum detectable sensitivity is -60 dBz down to 50 m altitude and the range resolution is 15–30 m and 1–3 s which can be adjusted by the user. Mira35 STSR (simultaneous transmission and simultaneous reception) cloud radar is used for cloud radar reflectivity, spectral width, SLDR, Doppler spectra, dual-wavelength ratio measurements and scans. The Mira35 minimum detectable sensitivity goes down to -50 dBz at 5 km height and range resolution is 30 m and 3 s. The radar can detect targets down to 150 m above ground, with full sensitivity above 450 m. CHM-15kx ceilometer data is going into the VOODOO algorithm. HATPRO G5 (Humidity and Temperature PROfiler Generation 5) microwave radiometer is used for LWP information. Its resolution is 100–1000 m and 1 s. For the VDPS method the scanning-mode data of Mira35 is used. If not stated differently, Cloudnet categorize files (categorize is the name of the files in Cloudnet) are used to calculate effective reflectivity, spectral width, SLDR, DWR, and used for retrievals. For peakTree and Doppler spectra raw data are used.

For in-situ measurements, data from the following instruments are used:

**Figure 2.** Schematic overview of instruments and methods which are described in this section and the physical process derived from these methods. In brackets the required measurement device is highlighted.

2DVD (2-dimensional video disdrometer) data are used for particle size distributions (PSD) and ice crystal number concentration (ICNC) calculations. The resolution of the 2DVD is 0.17 mm which corresponds to the smallest detectable particle size. VISSS (Video In Situ Snowfall Sensor) data are used for PSD and ICNC calculations. The resolution of the VISSS is 0.06 mm which corresponds to the smallest detectable particle size. Parsivel<sup>2</sup> 1D disdrometer data are used for precipitation measurements. Its resolution range is 0.5–25 mm. Windsondes S1H3 data are used for atmospheric profiles of meteorological parameters like temperature or relative humidity. Additional radiosonde data from a radiosonde launch in Payerne (46.8°N, 6.9°E) on 8 Jan 2024, 11 UTC is used to extend windsond measurements in Eriswil from the lowest 500 m above ground through the entire troposphere (GRUAN Lead Centre, 2024).

Further details on these instruments are provided in table 1. If not stated differently, all used data can be found in Ohneiser et al. (2025).

#### 3.2 ICON-D2 model data

135

Model data from the ICON-D2 forecasts are utilized, available from the DWD data center upon request (ICON-D2, 2024).

The ICON-D2 regional model, with a horizontal resolution of approximately 2.1 km, covers Germany, Switzerland, Austria,

and parts of neighboring countries. This high resolution allows for explicit simulation of atmospheric convection phenomena, such as thunderstorms, and enhances topographical representation, capturing more valleys and mountains, particularly in midmountain and Alpine regions.

ICON-D2 employs the single-moment microphysics scheme by Seifert (2008) to predict cloud water, rainwater, cloud ice, snow, and graupel. It provides new 48-hour forecasts every three hours, starting from 00 to 21 UTC. To assess model performance, precipitation rates and rime mass fraction are compared with observations from ground-based instruments and remote-sensing data. The latter is estimated from the ICON-D2 model as ratio of graupel content to total condensate content. ICON-D2 output is also used to compare atmospheric profiles of temperature, relative humidity and wind to radiosonde measurements.

# 155 3.3 Hybrid Single-Particle Lagrangeian Integrated Trajectory Model – HYSPLIT

HYSPLIT (Hybrid Single-Particle Lagrangeian Integrated Trajectory Model, Stein et al., 2015) is a model used to calculate trajectories, origin of air parcels, and dispersion. The model is freely available at HYSPLIT (2024). Meteorological data in terms of air pressure, temperature, humidity, wind speed, wind direction, or precipitation are required. The HYSPLIT model uses a Lagrangian approach as well as a Eulerian approach. The Lagrangian approach is used with a moving frame of reference for the advection and diffusion calculations. The Eulerian approach uses a fixed three-dimensional grid as a frame of reference to compute pollutant air concentrations. We use the standard HYSPLIT model. There are studies that show the influence of complex terrain on the accuracy of trajectories. Hernandez-Ceballos et al. (2014) show that the change from GDAS (Global Data Assimilation System) to WRF–ARW (Weather Research and Forecasting Model – Advanced Research WRF) enhances the accuracy of trajectories. Problems in complex terrain arise when meteorological forecast data is spatially averaged.

#### 165 3.4 Cloudnet centralized datasets

160

Comprehensive datasets require coordinated analysis schemes. Cloudnet (Illingworth et al., 2007; Tukiainen et al., 2020) is a tool that combines ground-based cloud remote-sensing measurements and model data in a harmonized, centralized, and quality assured structure focusing on long-term observations of clouds, aerosols, and precipitation. Therefore, the radar data is downloaded from Cloudnet. Data for 8 Jan 2024 is from Seifert and O'Connor (2025). The required data are from RPG94 cloud radar, Mira35 cloud radar, HATPRO G5 microwave radiometer, CHM-15kx ceilometer, and the weather model reanalysis data of ECMWF (European Centre for Medium-Range Weather Forecast ECMWF, 2024). All different radar variables are stored in a harmonized and user-friendly data structure in the so-called categorize files. The effective reflectivities are already provided in the categorize files from Cloudnet (Illingworth et al., 2007; Tukiainen et al., 2020; Seifert and O'Connor, 2025). In these files RPG94 data and Mira35 data is harmonized and stored quality assured. For this, the raw data is harmonized, for example into netCDF formats. Then calibration and quality control are done. The Level-2 cloud products generation is performed, including vertical profiles, cloud classification, and liquid/ice water content. Finally, harmonization steps like bias correction, inter-site instrument matching, and consistency checks are done. All processing steps are traceable and reproducible and the datasets get an individual DOI.

# 3.5 Fall streak tracking algorithm

Existing fall streak tracking algorithms, such as those developed by Kalesse et al. (2016b), Pfitzenmaier et al. (2017), and Pfitzenmaier et al. (2018), focus on identifying generating cells and determining the precipitation initiation temperature. These algorithms use three-dimensional wind information to calculate fall streaks, which works effectively in conditions without wind shear. However, when wind shear is present, the accuracy of these approaches is limited. A common alternative is to define a fall streak based on local maxima in radar reflectivity, as demonstrated by Browne (1952) and Marshall (1953) and then also used by Ramelli et al. (2021a) and Ramelli et al. (2021b), among others. This method has been adopted in this study as well. The wind-based fall streak approach is usually prioritized if precise wind data with a high resolution are available or the reflectivity field is weak or noisy. In contrast, the reflectivity-based approach is used if reflectivity features are directly associated with hydrometeor descent, or when there is no high-quality wind data available. Presence of directional wind shear remains the largest source of uncertainties.

The algorithm in this study (see Ohneiser, 2025) first requires a time of interest at the lowest radar range bin, ensuring that this is the point at which the fall streak arrives to have the best comparability with ground-based in-situ measurements. It then searches for the local maximum in reflectivity in the next higher radar height bin within a  $\pm 30$  second window. This process is repeated iteratively until cloud top is reached, reconstructing the pathway of the fall streak. The resulting trajectory provides a continuous view of the microphysical changes occurring along the fall streak.

While this approach is an improvement over simple vertical profiles, it is not without limitations. In presence of wind shear, or when reflectivity gradients are weak, the algorithm may struggle to accurately trace the fall streak. Additionally, real atmospheric conditions often introduce three-dimensional effects that cannot be fully captured using two-dimensional radar observations (time-height cross-sections). Despite these challenges, this approach offers a more realistic representation of ice crystal trajectories compared to vertical profiles alone, and thus represents the best possible assumption under the given circumstances. Also, it needs to be mentioned that it tracks the population having the largest contribution to  $Z_e$ , and smaller particles may have different trajectories in the cloud. In the following sections, the fall streak tracking algorithm will be applied to radar data (RPG94) to analyze the observed cloud systems.

# 3.6 VOODOO

The detection of supercooled liquid layers in clouds is important to estimate the effect of riming. With cloud radars alone, it is hard to detect supercooled liquid layers in a cloud as ice crystals always dominate the magnitude of the reflectivity. The combination of radar and lidar enables detection of liquid layers in a cloud. However, if the lidar signal is attenuated within an optically thick cloud, liquid layers can be missed. VOODOO is a retrieval based on cloud radar Doppler spectra, enabling the detection of liquid layers throughout the cloud.

It is based on deep convolutional neural networks mapping Doppler spectral characteristics from vertically-pointing cloud radar observations to the probability of the presence of cloud droplets. All details about the algorithm can be found in Schimmel et al. (2022). VOODOO makes use of CLOUDNET data.

VOODOO utilizes cloud radar Doppler spectra from a vertically pointing radar, attenuated backscatter coefficient at 1064 nm from a ceilometer, liquid water path (LWP) retrieved from a microwave radiometer, and temperature, relative humidity, and pressure from numerical weather forecast data from the European Centre for Medium-Range Weather Forecasts (ECMWF). The grid size in VOODOO is adjusted to the one of Cloudnet, so the temporal resolution is 30 s and the range resolution is between 30 and 45 m, depending on radar settings.

In radar Doppler spectra, distinct peaks are a result of different terminal fall velocities of different particle habits. Ice crystals have a comparably larger terminal fall velocity. Liquid water, however, would show a small peak in reflectivity at a terminal fall velocity around  $0 \,\mathrm{m\,s^{-1}}$ . In this way, liquid water is detected beyond lidar attenuation. The technique also has its limitations because turbulence adds difficulties to find a liquid peak at  $0 \,\mathrm{m\,s^{-1}}$ . In addition, the reflectivity can be too low and the liquid peak can be overlapped with ice crystal signals. According to Schimmel et al. (2022), VOODOO performs best for (multi-layer) stratiform, deep mixed-phase cloud situations.

#### 3.7 Dual-wavelength ratio (DWR) technique

For hydrometeors whose size is approximately a factor of 5 smaller than the wavelength of observations, the backscattering efficiency at 35 GHz and 94 GHz is different. This fact can be used to gain additional insights into cloud microphysical processes. Using the ratio of radar reflectivities at these two radar wavelengths gives information about the size of hydrometeors. The dual-wavelength ratio (DWR) increases with particle size when the shorter radar wavelength is equal to or shorter than the particle size (Tetoni et al., 2022; Chellini et al., 2022). High DWR values can thus indicate the presence of liquid water or rimed or aggregated ice crystals.

The dual wavelength ratio (DWR<sub>35/94</sub>) is based on radar reflectivities between Mira35 and RPG94. To achieve this, several calculation steps are required. First, the radar reflectivities must be corrected for two-way radar attenuation caused by atmospheric gases. The effective reflectivities are already provided in the categorize files from Cloudnet (see Sect. 3.4). Next, it is essential to interpolate the data in such a way that both radar systems are aligned on the same time and height grid, e.g. to the grid of the RPG94 cloud radar. For example, the data with the fine resolution can be transferred to the coarse resolution, so that both datasets can be subtracted from each other. The DWR is then calculated following the method of Matrosov et al. (2022), using the difference in effective reflectivity (in logarithmic scale) between the two radar systems:  $Z_{e,35}$  for the Mira35 cloud radar and  $Z_{e,94}$  for the RPG94 cloud radar:

$$DWR_{35/94} = Z_{e,35} - Z_{e,94} \tag{1}$$

The DWR calibration is performed at cloud top, where it is assumed that DWR<sub>35/94</sub> = 0. The size of particles at cloud top has not been evaluated. However, cloud tops typically consist of smaller ice crystals because the cloud top is prone to new particle formation. Therefore, we assume the particles are small in this range. When ice crystals are small enough they are closer to the Rayleigh scattering regime and the difference in scattering between 35 GHz and 94 GHz is minimal, resulting in a near-zero DWR and thus preferable conditions for calibration of the DWR. To reduce noise, the uppermost five range rates are taken into

account. This means that the average DWR<sub>35/94</sub> obtained at the uppermost five range gates cloud top is subtracted from the DWR profile at each time step. Enhanced DWR can indicate processes like riming or aggregation but also liquid attenuation will enhance the DWR.

# 3.8 Eddy dissipation rate retrieval

250

260

265

The eddy dissipation rate (EDR) is a measure of atmospheric turbulence. The EDR defines the rate at which turbulent kinetic energy cascades from large to small eddies in the atmosphere until energy is converted from mechanical into thermal energy at molecular level (Foken, 2008). Within updrafts, liquid droplet formation can be enhanced. A large EDR corresponds to quickly dissipating energy and that the atmospheric turbulence level is high. In addition, higher turbulence also results in a stronger interaction between all hydrometeors in a cloud, potentially enhancing riming and aggregation efficiency. It serves as an indicator for turbulence. The method for estimating EDR using Cloudnet horizontal and vertical wind speeds from ECMWF, along with radar mean Doppler velocity (MDV), was developed by Frisch and Strauch (1976); Shupe et al. (2012); Borque et al. (2016) and adjusted by Griesche et al. (2020). More information on the calculation of EDR can also be found in Vogl et al. (2024).

# 3.9 peakTree: tool for cloud radar Doppler spectrum peak analysis

Cloud radar observations frequently contain information on multiple ice crystal species in the observation volume when there are distinct peaks in the Doppler spectrum. The algorithm peakTree is designed to analyze the peaks inside such a Doppler spectrum and to separate the contribution of different particle populations to the Doppler spectrum, such as liquid cloud droplets, drizzle, rain and various ice crystal habits. It uses a recursive approach, representing subpeaks as nodes in a binary tree structure to represent the peaks of the radar Doppler spectrum. If the reflectivity is above the noise floor, all peaks that have a prominence of more than 1 dB are classified as a subpeak. Further details are provided in Radenz et al. (2019) as well as in Vogl et al. (2024). Each clearly separated Doppler peak corresponds to a particle mode, for example liquid cloud droplets, needles, dendrites, or other.

Cloud droplets are small and have a negligible fall velocity, so they can be assumed to be a proxy for air motion. As a consequence, the difference between the slowest falling  $v_{\min}$  (i.e. liquid cloud droplets) and fastest falling hydrometeor habit  $v_{\max}$  (snow flakes or graupel) can be considered as the vertical-air-motion-corrected fall velocity of the fast falling hydrometeor. As a threshold,  $1.5\,\mathrm{m\,s^{-1}}$  is used to discriminate heavily rimed ice crystals (graupel or hail) from unrimed snowflakes. Even the largest aggregates would not reach these velocities, so that this threshold assures that only graupel particles are found in the fast falling branch.

Based on the Doppler spectrum sub-peak moments derived by peakTree, a hydrometeor classification is applied in cloud regions classified as "ice" or "ice and liquid" by Cloudnet. The hydrometeor classification relies on following thresholds:

Supercooled liquid cloud droplet peaks are defined as slow-falling sub-peaks with low reflectivity, having low mean Doppler velocity: |MDV| < 0.3 and  $Z_e < -10$  dBZ.

Columnar ice peaks exhibit low Z<sub>e</sub> and low MDV, but are additionally characterized by enhanced SLDR. As SLDR for sub-

peaks is not available in peakTree yet for STSR radar processing, we rely on a threshold applied to the SLDR of the entire spectrum instead: A spectrum is classified as "columnar ice-containing" if it contains a peak with  $Z_{\rm e} < 5\,{\rm dBZ}$  (see Oue et al., 2015, for high numbers of columnar ice) and SLDR of the entire spectrum is higher than  $-20\,dB$ . Table 2 gives more detailed information on the thresholds used for the classification.

Rimed ice peaks are defined as peaks falling at least 1.5 m s<sup>-1</sup> faster than another, slower-falling sub-peak in the same spectrum. Following this approach, some Doppler spectra in regions classified as "ice" or "ice and liquid" fulfill more than one of the above criteria, while many spectra do not fulfill any. Spectra in which none of the sub-peaks meet the classification thresholds are labeled as "unclassified". These pixels likely correspond to larger ice particles, or small ice that does not lead to increased SLDR signatures in the vertically-pointing radar. In cases where two or more sub-peaks satisfy different classification criteria (e.g. both "supercooled liquid droplets" and "rimed"), a mixed class is assigned, such as "liquid droplets and rimed ice".

Table 2. Peak selection criteria for peakTree-based hydrometeor classification

# Hydrometeor Class Selection Thresholds liquid droplets

- $Z_{\rm e} < -10 \ {
  m dBZ}$
- |MDV| < 0.3 m/s

# for STSR radar:

- Retained only if bulk LDR  $\leq$  -20 dB, or if at least one additional peak with  $Z_{\rm e}$  < 5 dBZ exists

#### columnar ice peak

–  $Z_{\rm e} < 5~{
m dBZ}$ 

#### for STSR radar:

- Spectrum bulk LDR > -10 dB

#### for LDR mode radar:

peak LDR > -20 dB

# rimed ice particles

– node falling  $\geq 1.5$  m/s faster than another, slower-falling node in the same spectrum

# 3.10 Ice crystal shape retrieval based on scanning cloud radar observations

The VDPS (vertical distribution of ice crystal shape, Teisseire et al., 2025) method aims to characterize the shape of cloud particles from SLDR-mode (slanted linear depolarization ratio) scanning cloud radar observations. This approach combines values from a scattering model developed by Myagkov et al. (2016) and measurements of SLDR at different elevation angles  $\Theta$ . The spheroidal scattering model delivers polarizability ratio ( $\xi$ ) and degree of orientation ( $\kappa$ ), which describe the apparent ice crystal shape by means of a density-weighted axis ratio and their preferred orientation, respectively. A polarizability ratio  $\xi \approx 1$  (when the polarizability ratio  $\xi$  takes values between 0.8 to 1.2) corresponds to isometric ice crystals characterizing spherical ice crystals or ice crystals with low density. On the contrary, a polarizability ratio  $\xi < 0.8$  and  $\xi > 1.2$  hints at oblate and prolate ice crystals, respectively. The VDPS method is described in detail in Teisseire et al. (2025), where the approach is validated by means of three case studies representing the three primary ice crystal shape classes, prolate, isometric and oblate ice crystal shapes, and where it is applied to discriminate riming and aggregation processes (Teisseire et al., 2024).

#### 3.11 Riming retrieval based on radar data

The seeding of ice crystals into a supercooled liquid water layer is likely related to riming. Riming enhances mass, size and fall velocity of ice crystals and changes it towards more spherical shapes. We are using a riming retrieval based on an artificial neural network taking ground-based, zenith-pointing cloud radar variables as input features (Vogl et al., 2022). Here,  $Z_e$ , MDV, the width from left to right edge of the cloud radar Doppler spectrum above noise floor, and skewness are used from the RPG94 cloud radar. An artificial neural network is then applied to predict riming. Details of the retrieval can be found in Vogl et al. (2022). If the MDV is equivalent to the particle fall speed in the observation volume the method is not reliable. Also, the spatiotemporal mismatch of radar and PIP (Precipitation Imaging Package) observations can add errors to the results.

#### 3.12 Riming retrieval based on VISSS data

In-situ measurements of precipitation can also yield information about riming. The extended observations during the field campaign thus offer an ideal opportunity to validate the accuracy of remote-sensing retrievals against in-situ measurements. The calculation of the sum of rimed mass  $(m_{\text{rime}}^{\text{sum}})$  is derived from the normalized rime mass  $(M_i)$  defined as the rime mass  $(m_{\text{rime}})$  normalized by the mass of size-equivalent graupel particle  $(m_g)$  (Garrett and Yuter, 2014).

$$M_{\rm i} = \frac{m_{\rm rime}}{m_{\rm g}} = \frac{m_{\rm rime}}{\frac{\pi}{6}\rho D^3},\tag{2}$$

where the graupel density ( $\rho$ ) is assumed to be 700 kg m<sup>-3</sup> (Seifert et al., 2019), and D represents the maximum particle size. Here, the in-situ method to derive  $M_i$  from Maherndl et al. (2024) is used to quantify  $M_i$  for individual particles from VISSS measurements of particle cross-sectional area, maximum size, and perimeter. Details of this retrieval are described in Maherndl et al. (2023, 2024). Only particles with a maximum size larger than 20 pixels corresponding to D > 1.18 mm are used to derive

 $M_{\rm i}$ . This is done because  $M_{\rm i}$  is biased for small particles, which all have round shapes due to limits in the resolution of VISSS which is 0.06 mm.  $m_{\rm rime}^{\rm sum}$  is derived by calculating the sum of  $m_{\rm rime}$  over particles with D > 1.18 mm for one-minute intervals. M is estimated for the particle population by calculating the average of  $M_{\rm i}$  of all particles with D > 1.18 mm in one-minute intervals. The fraction of rimed mass (FR) for a particle population is computed using the formula adapted from Kneifel and Moisseev (2020), Maherndl et al. (2023), and Maherndl et al. (2024):

$$FR = MD^{3-\beta_{\rm m}} \frac{\pi \rho}{6\alpha_{\rm m}} \tag{3}$$

In this formula,  $\beta_m$  and  $\alpha_m$  are the exponents and prefactors of the mass-size relation, both of which depend on M (Maherndl et al., 2023).

While FR presents a measure of fraction of rime mass to total snow particle mass,  $m_{\text{rime}}^{\text{sum}}$  quantifies the total rime mass. Assuming a rimed particle population,  $m_{\text{rime}}^{\text{sum}}$  depends on the total precipitation amount, while FR does not.

#### 3.13 ICNC retrieval based on 2DVD data

With in-situ measurements of precipitation by the 2DVD it is possible to obtain information about particle number concentration (PNC). If all particles are ice crystals, the PNC is equal to the ice crystal number concentration (*ICNC*).

The *ICNC* in m<sup>-3</sup> depends on measurement time  $\Delta t$ , effective measuring area  $A_{\rm eff}$ , vertical velocity  $v_{\rm j}$  of the particle j, and number of measured particles M. The formula for calculating ICNC using 2DVD data (see Gaudek, 2024) results to:

$$ICNC = \frac{1}{\Delta t} \sum_{j=1}^{M} \frac{1}{A_{\text{eff,j}} \cdot 10^{-6} \cdot v_{\text{j}}}$$
 (4)

For the calculation of the particle size distribution PSD in m<sup>-3</sup> mm<sup>-1</sup> it is necessary to define a size class width  $\Delta D$  with the size class i:

$$PSD = \frac{1}{\Delta t \Delta D} \sum_{j=1}^{M_{\rm i}} \frac{1}{A_{\rm eff,i,j} \cdot 10^{-6} \cdot v_{\rm i,j}}$$
 (5)

#### 4 Synoptic situation in Eriswil, Switzerland, on 8 Jan 2024

The Swiss Plateau is known for its persistent low-level stratus clouds, particularly during a weather pattern called "Bise" situation (Granwehr, 2022). Bise typically occurs in winter when a high-pressure system is located over northern or central Europe, while a low-pressure system is positioned over southern Europe. In this setup, cold and moist air is transported towards central Switzerland by northeasterly winds in the lowest approximately 2 km of the atmosphere. This leads to the formation of persistent low-level stratus clouds, with temperatures at the top of the cloud (around 2 km above ground) generally ranging between  $-2^{\circ}$ C and  $-8^{\circ}$ C.

Figure 3a shows 72-hours backward trajectories of air masses arriving at heights of 500, 1900, and 4500 m a.g.l. at Eriswil on

8 Jan 2024, 4 UTC. The near-surface air mass originates from northeastern Europe and traveled across Poland and Germany. A slightly drier air mass, located above, approaches from central eastern Europe. The uppermost air mass originates from northern Africa, passing over the Mediterranean Sea and Italy. According to CAMS data (see Fig. 3d), this air mass contains Saharan dust. More than half of the aerosols are related to dust. Additionally, it is influenced by remnants of an occluded frontal system moving northward across Switzerland (not shown), which contributes to cloud formation in the free troposphere and precipitation.

Figure 3b presents atmospheric temperature and relative humidity profiles from ICON-D2, windsondes and radiosondes. According to windsond measurements that are only available in the lowest few hundred meters of the atmosphere launched at Eriswil, the surface temperature is around –8°C, while the cloud-top temperature of the Bise cloud, approximately 1 km above ground, is about –10°C. The ICON-D2 forecast indicates a temperature inversion close to the ground (around 1 km above ground, see Fig. 3b). Above the inversion, with increasing altitude, temperatures decrease up to the tropopause, located at around 9 km height. For comparison, also the radiosonde launch at Payerne is shown. In the lowest 1 km of the atmosphere, the data fits well to the windsond launch in Eriswil, so the estimate for the entire atmosphere is probably realistic. It shows difficulties that ICON-D2 has with the representation of the inversion disconnecting the planetary boundary layer from the free troposphere. The relative humidity is high throughout the entire troposphere. Windsond data shows that the lowest kilometer of the atmosphere above Eriswil is nearly saturated with respect to liquid water. The higher altitudes are also almost saturated w.r.t. liquid water or even supersaturated with respect to ice, according to the ICON-D2 forecast. The slightly drier layer between 1 and 2.5 km height is clearly visible in the radiosonde data from Payerne.

As illustrated in Fig. 3c, northeasterly winds prevail in the lowest, saturated air mass. Above the inversion layer, winds shift to southeasterly and southerly directions. The strongest winds of about  $9-10 \,\mathrm{m\,s^{-1}}$  are observed near the surface and between 6 and 10 km altitude.

We checked the aerosol situation with CAMS data. With northeasterly winds at the surface, the total aerosol load is very high in the PBL. On top of the inversion the aerosol load is much lower. The dust load, however, is lower within the PBL (due to northeasterly winds) while in the free troposphere, southerly winds prevail and dust load increases. This high dust fraction must also be considered as a potential INP source for the seeder cloud in the free troposphere.

#### 370 5 Observations

#### 5.1 Case study 8 Jan 2024 – a remote-sensing and in-situ overview

This case study provides an ideal scenario to compare three different seeding conditions: 1) Strong seeding: Ice crystals that are minimally affected by sublimation in the dry layer. In that scenario, the radar reflectivity between the cloud layers does not decrease significantly, indicating that a large ice mass reaches the lower cloud that consists of mostly supercooled liquid water but also a few ice crystals, likely resulting in strong riming. 2) Weak seeding: Ice crystals that are partially or mostly sublimated in the dry layer. The remaining ice mass was relatively small, and crystals that reach the lower cloud likely contain

**Figure 3.** Meteorological situation on 8 Jan 2024 over Eriswil. a) HYSPLIT backward trajectories ending at Eriswil on 8 Jan 2024, at 04 UTC at 500, 1900, and 4500 m a.g.l.; b) Temperature from ICON-D2 model ( $T_{\rm ICON}$ ), windsond ( $T_{\rm RS,Eri}$ ) at Eriswil and radiosonde ( $T_{\rm RS,Pay}$ ) at Payerne (46.82°N 6.94°E) and relative humidity over water (windsond Eriswil:  $RH_{\rm RS,Eri,w}$ , radiosonde Payerne:  $RH_{\rm RS,Eri,w}$ , ICON:  $RH_{\rm ICON,w}$ ) and over ice (ICON:  $RH_{\rm ICON,i}$ ). Windsond data in Eriswil in b) is from 8 Jan 2024, 12:31 UTC; Radiosonde data in Payerne is from 8 Jan 2024, 11 UTC c) wind speed and direction from ICON-D2. The direction of the arrows indicates the wind direction. Visualized data in b) and c) are both from 8 Jan 2024, 9 UTC from the ICON model run.

many small fragments from incomplete sublimation. 3) No seeding: The lower supercooled liquid cloud (also containing a few ice crystals) is not affected by seeding from the upper cloud. This results in lower ice production and radar reflectivity at cloud base compared to seeding scenarios.

Figure 4a shows the temperature profile from the radio sounding at Payerne on 8 Jan 2024 at 11 UTC. The surface temperatures w.r.t. Eriswil were around –8°C and the inversion was at around 1.2 km above ground. Above the inversion, the temperature decreases until the tropopause which is located around 9 km above ground.

Figure 4b displays the radar reflectivity from the RPG94 cloud radar. It provides an overview of the analyzed scene in this study. Clouds are present throughout the entire period. In the feeder cloud the reflectivity was between 5 and 20 dBZ during seeding and between –5 and 5 dBZ without seeding. Notably, the Bise cloud remained liquid (with only a negligible amount of ice crystals) throughout the entire considered period, during the phase of strong interaction with the seeder cloud until approximately 9 UTC, as well as afterwards, when the two clouds are decoupled.

The seeder cloud layer produces ice crystals that fall towards the feeder cloud. Between 0 and 9 UTC, some ice crystals do not

fully sublimate in the drier layer between the cloud systems at around 1.5 to 4 km altitude but fall into the supercooled stratus cloud. This results in an increase in radar reflectivity within the lower cloud to values between 5 and 20 dBZ (compared to values between –5 and 5 dBZ in the feeder cloud after 9 UTC). With the INP sampler Digitel DPA-14 it is observed that there is a lack of INP in the planetary boundary layer (not shown here), so the supercooled Bise cloud does not produce significant amounts of ice and has therefore a very high liquid water path up to 300–400 g m<sup>-2</sup>. It can be assumed that this high availability of liquid water leads to the dramatic increase in radar reflectivity because of ice crystal growth at this height.

In contrast, after 9 UTC, ice crystals from the upper cloud completely sublimate in the drier layer, stopping further seeding. As a result, there is a clear separation between the cloud systems and reflectivity in the low-level stratus cloud decreases to values between –5 and 5 dBZ.

The fall streak tracking algorithm is applied to this case study to get an idea on microphysical changes of hydrometeors along their trajectories. As stated in the methods section, it is physically not possible to follow the actual hydrometeors of interest and also the fall streak follows the population that contributes most to  $Z_e$ . Nevertheless, it is the best possible estimate for the ongoing changes within the cloud system. Of course wind shear is generally low within the seeder (southerly winds) and feeder (easterly winds) cloud. Wind direction shear only occurs in the zone between the two cloud layers. This means that the algorithm is more reliable within the seeder cloud and within the feeder cloud but less reliable in the interaction region.

Figure 4c shows the reflectivity along fall streaks for the three seeding categories. The purple curves (strong seeding) reveal an almost or at least in large parts of the cloud a steady increase in reflectivity from cloud top to base. The orange curves (weak seeding) are also characterized by the same increase in reflectivity until approximately 2 km a.g.l., where it abruptly decreases due to sublimation. Nevertheless, a connection to the lower cloud remains (criterion for weak seeding) and a strong increase in reflectivity from around –25 dBZ up to 10 dBZ is observed below 2 km. At the lowest radar range bin (around 119 m above ground), reflectivity values for strong and weak seeding cases are similar. The blue curves (Bise cloud only) show cloud tops around 1 km above ground with much lower reflectivity values between –5 dBZ and 5 dBZ at cloud base compared to the seeding cases.

The LWP measurements that is shown in Fig. 11a) show that the LWP was typically between 100 and  $300 \,\mathrm{g}\,m^{-2}$ . It decreased while strong seeding took place. The stable background of LWP was around  $300 \,\mathrm{g}\,m^{-2}$  and quite constant during the time where no seeding took place (around 9–16 UTC). The decrease in LWP during seeding can be attributed to the riming process.

Figure 5 shows an analysis of vertically pointing cloud radar observations at 94 GHz. The derived quantities of spectral width, slanted linear depolarization ratio (SLDR), probability of cloud droplet presence, rimed mass fraction, dual wavelength ratio (of 35 GHz and 94 GHz radar) and EDR explain the microphysical properties of the cloud system.

Figure 5a presents the time-height cross-section of the Doppler spectral width. The seeder cloud is characterized by low spectral width values, indicating a predominantly monomodal ice crystal size distribution, composed of ice crystals. However, at around 2 to 3 km height, increased values of spectral width are found. It suggests the presence of coexisting particle habits. An increased spectral width can also be caused by wind shear or turbulence. Either columns and aggregates or columns and supercooled water droplets coexist. In the feeder cloud, below 1 km, the spectral width is significantly enhanced, reaching up

**Figure 4.** a) Radio sounding at Payerne (46.82°N 6.94°E) on 8 Jan 2024, 11 UTC. Data is from GRUAN Lead Centre (2024). b) Overview of the seeding event on 8 Jan 2024, 0–16 UTC. The RPG94 reflectivity is shown as a time-height cross-section. The colored lines from cloud top to cloud base indicate fall streaks. The different colors indicate strong seeding (violet), weak seeding (orange), and no seeding (blue). The seeder cloud is located at around 1.5 km to 8 km above ground. The feeder cloud is located from the lowest radar range bin to around 1.5 km height. c) The reflectivity along fall streaks shown in a).

to 0.5 m s<sup>-1</sup>. This broadening of the Doppler spectrum is likely caused by interaction of ice crystals from the seeder cloud with supercooled liquid water in the feeder cloud. This is also corroborated by VOODOO results (see below). Turbulence likely contributes to further widening of the spectrum.

Within the seeder cloud, SLDR values (shown in Fig. 5b) range between –20 dB and –25 dB, indicating that ice crystals are the dominant particle type. Additionally, before 9 UTC an enhancement of SLDR is observed from the top of the Bise cloud down to the surface, suggesting coexistence of supercooled water and ice crystals. After 9 UTC, SLDR in the Bise cloud decreases to around –28 dB, indicating predominantly spherical hydrometeors, likely supercooled liquid water. It should be noted that SLDR increases by a few dBZ on the way from the top of the Bise cloud downwards.

Figure 5c illustrates the probability of cloud droplet presence, as retrieved by VOODOO. The seeder cloud shows a relatively low probability, indicating predominantly ice crystals, as also noted in Figure 5a. Below 1 km and before 9 UTC, the probability of cloud droplet presence is highest, pointing to coexistence of liquid water and ice. After 9 UTC, this probability decreases slightly.

Figure 5d displays the rimed mass fraction, derived from the retrieval method of Vogl et al. (2022). No riming features are observed in the upper part of the upper cloud. However, between 3 and 3.8 km height, increased probability of riming is visible. However, the remote-sensing technique is at its limit because in the end two options remain. It could be a result of supercooled liquid water in this part of the cloud or increased turbulence due to latent heat release. The combination of increased spectral width (Fig. 5a), increased riming fraction (Fig. 5c) and increased probability of liquid water to be present through VOODOO (Fig. 5d) leaves the question open if supercooled liquid water, aggregates or a combination of both was present in the region

**Figure 5.** Radar products and retrievals of RPG94 on 8 Jan 2024, 0–16 UTC. a) shows the spectral width. b) shows the slanted linear depolarization ratio (SLDR). Grey color indicates regions where there is a cloud signal in the co-channel, but the signal in the cross-channel is not sufficient to calculate SLDR. c) shows the probability of cloud droplet occurrence retrieved with VOODOO. Values below 40% were set to grey colors to highlight regions with increased probability of cloud droplets. d) shows the radar-retrieved rimed mass fraction. Values below 40% were set to grey colors to highlight regions with increased probability of riming. e) shows the dual-wavelength ratio calculated between Mira35 and RPG94 cloud radars. In f) EDR determined from 94 GHz radar observations is shown.

of 3.0 to 3.8 km height of the cloud. To answer this question, PAMTRA (Passive and Active Microwave radiative TRAnsfer; Mech et al., 2020) simulations are conducted to get an estimation by how much the DWR would be increased if liquid water would be present. The presence of liquid water would increase the DWR because of stronger attenuation at 94 GHz compared to 35 GHz in the supercooled liquid water layer.

In the observations, a significant enhancement of 2–4 dB in the dual wavelength ratio is visible. For PAMTRA, a layer of ice particles with a reflectivity of  $-10 \, \text{dB}$  is used as a height-constant background between 3.0 and 3.8 km height. In the following calculations, a supercooled liquid water droplet layer is added to this layer. For a liquid contribution of  $0.15 \, \text{g m}^{-3}$ ,  $0.35 \, \text{g m}^{-3}$ , and  $0.50 \, \text{g m}^{-3}$  (corresponding to a LWP of 120, 280, and  $400 \, \text{g m}^{-2}$ , respectively) DWR would increase by  $0.6 \, \text{dB}$ ,  $1.3 \, \text{dB}$ , and  $1.9 \, \text{dB}$ , respectively. These numbers are consistent with findings in Lebsock et al. (2011).

As can be seen in Fig. 11 LWP was around  $300 \,\mathrm{g}\,\mathrm{m}^{-2}$ , and a large percentage was contributed by the Bise cloud as such. The observed increase in DWR could even not be reproduced with PAMTRA when assuming  $0.5 \,\mathrm{g}\,\mathrm{m}^{-3}$  (corresponding to  $400 \,\mathrm{g}\,\mathrm{m}^{-2}$ ). To conclude, this would mean that liquid water is probably not the dominating second particle type. It is more likely that aggregation is happening at this height range.

As ice crystals fell into the Bise cloud, the rimed mass fraction increased abruptly, exceeding 70%, due to the high availability of liquid water. In contrast, after 9 UTC, when seeding no longer occurs, the rimed mass fraction dropped significantly.

Figure 5e shows the dual-wavelength ratio between the Mira35 and RPG94 cloud radars. In the seeder cloud, DWR remains close to 0 from cloud top down to approximately 3–4 km height. Before 9 UTC, increases in DWR between 1.5 and 3 km might suggest ice crystal aggregation and/or riming as discussed before. A more pronounced DWR increase occurs in the feeder cloud during seeding, with values reaching 5–10 dB, which could be due to significant riming, steady particle growth, as well as coexistence of liquid water. After 9 UTC, DWR in the feeder cloud decreases to around 0 dB, reflecting the absence of seeding and the lack of riming or aggregation processes towards large particle sizes. This is also the region with the highest values of EDR, shown in Fig. 5f. Higher values correspond to stronger turbulence. The highest turbulence is observed within the Bise cloud throughout the period, with a tendency for increased turbulence in the morning during seeding. Slightly enhanced turbulence is also seen at the seeder cloud top, while turbulence within most parts of the seeder cloud is small.

Figure 6 presents results of 2DVD and VISSS measurements. Figures 6a and b compare the particle size distributions from both instruments. In principal, both measurement devices should show the same results, however, VISSS can detect even smaller particles than the 2DVD which are below 0.5 mm in diameter. Also, 2DVD seems to detect a few more larger particles larger than 4 mm. This could also be an artifact caused by the particle detection algorithm. Both datasets show that particle number and size are larger during the seeding phase in the morning compared to after 9 UTC. Exemplary VISSS images in Fig. 6e highlight the presence of larger, more complex-shaped and strongly rimed particles between 0 and 8 UTC, contrasting with smaller, nearly spherical, ice crystals observed after 9 UTC. These smaller particles have a peak number concentration at around 1 mm in size, visible with orange colors in Fig. 6a and b. The high numbers in the smallest bin sizes can be accounted for as an artifact because these are closest to the detection limit for both instruments. This contrast is further reflected in the retrieved ice crystal number concentration (ICNC) shown in Fig. 6c and d, where higher ICNC values of up to 70–100 L<sup>-1</sup> were recorded during the seeding period, compared to less than 5–50 L<sup>-1</sup> after 9 UTC. This decrease in particle number fits well to

**Figure 6.** 2DVD and VISSS observations are shown. a) and b) particle size distributions of VISSS and 2DVD, respectively, are shown as number concentrations. In c) and d) the ICNC algorithm is applied to VISSS and 2DVD data, respectively. The gray rectangles indicate time periods with no available data because of problems with the data acquisition. In e) quicklooks showing a random selection of particles observed by VISSS are shown from 3:23–3:59 UTC and 12:07–12:14 UTC.

findings in Fig. 5 and Fig. 8 that show a larger number and size of particles during seeding compared to without seeding. The absolute numbers between VISSS and 2DVD data seem to be a bit different. VISSS detects more ice crystals than the 2DVD. As emphasized before, VISSS detects more of the very small particles below 0.5 mm which might explain differences in the

# 5.2 Microphysical analysis of seeder-feeder cloud interactions using radar observations

**Figure 7.** a) Doppler spectrogram measured with the Mira35 cloud radar of a case with strong seeding on 8 Jan 2024 at 04:28 UTC. The Doppler spectrum in b) is from 1 km height from the case in a) with the noise level as determined by the Mira-35 processing software by using a Hildebrand-Sekon noise level detection technique (Görsdorf et al., 2015).

Figure 7a presents a Doppler spectrogram during a strong seeding event at 4:28 UTC. This is during a scan period of Mira35, so no fall streaks were available during this time. The absolute Doppler velocity increases gradually along the ice crystal pathway from cloud top to cloud base, reflecting growth and acceleration of ice crystals as they fall toward the Bise cloud. At the top of the Bise cloud, two additional peaks appear in the Doppler spectra, suggesting presence of three particle populations. Larger and faster falling ice crystals might be co-located with smaller and slower falling ice crystals from another fall streak and these interact with the Bise cloud (which is characterized by Doppler velocities around 0 m s<sup>-1</sup>) in addition. In Fig. 7b, a Doppler spectrum with three well-separated peaks is shown at 1 km altitude. The slowest-falling (right) peak corresponds to supercooled cloud droplets from the Bise cloud because the Doppler velocities are around 0 m s<sup>-1</sup>. This is a typical indication for liquid water droplets. The faster falling two peaks (left side of the spectrum) are ice crystals originating from the seeder cloud and probably also from the interaction of the seeder and feeder cloud or could be the result of locally-generated ice crystals. The central peak is associated with the highest reflectivities, probably indicating its significant contribution to overall precipitation.

Figure 8 provides a more detailed analysis of the number of peaks in individual Doppler spectra using the peakTree algorithm.

One of such individual Doppler spectra is shown in the example of a Doppler spectrogram and Doppler spectrum at 1 km height

at 4:28 UTC in Figure 7b. Figure 8a displays the number of modes detected. The general scheme of how a binary tree is set up including nodes and modes can be found in Radenz et al. (2019) and is also described in Sect. 3.9. The seeder cloud typically contains one mode and occasionally two modes. However, in the transition zone between seeder and feeder clouds, around 1 km above ground level, two and three modes are more common. This indicates that supercooled droplets are interacting with seeding ice crystals, leading to formation of additional modes, which supports the observations in Fig. 7b. Also the increase in rimed mass fraction and DWR shown in Fig. 5 support the coexistence of liquid water and ice. The high probability of liquid cloud droplets in the Bise cloud is also supported by peakTree results in Fig. 8, VOODOO results in Fig. 5c and the slowest falling Doppler spectrum peak in Fig. 7b.

Figure 8b offers a more in-depth analysis of the peakTree results, focusing on the difference between Doppler velocity of the fastest and slowest node in each spectrum. In the Bise cloud, it can be assumed to be the best approximation that the slowest-falling particles are typically supercooled cloud droplets, which have velocities similar to air motion. Therefore, the difference  $v_{\min} - v_{\max}$  can to some extent be interpreted as a rough estimate of turbulence-corrected fall velocity of snow crystals, at least within the Bise cloud.

In Figure 8b, during seeding, the fastest fall velocities reach up to  $1.8 \,\mathrm{m\,s^{-1}}$  within the Bise cloud, whereas, without seeding,

**Figure 8.** a) Number of modes in each of the Doppler spectra of Mira35 data after applying the peakTree algorithm. Vertical white lines are periods with no data because of ongoing scans. b) Difference in fall velocities of the fastest and slowest falling particles. c) zoomed version of b) for 2–6 UTC and 0–1.8 km height. d) peakTree-hydrometeor classification based on RPG94-GHz cloud radar data.

after 9 UTC, they only reach about  $0.5 \,\mathrm{m\,s^{-1}}$ . This difference suggests that, in the absence of seeding, ice crystals do not grow

significantly in this region. A steady increase in fall velocity, as shown more clearly in the zoomed view in Fig. 8c, strongly indicates the occurrence of riming or aggregation. This process increases size and density of particles, causing them to become heavier and fall faster (Kneifel and Moisseev, 2020). Terminal fall velocities exceeding 1.5 m s<sup>-1</sup> can be considered as a threshold for identifying graupel (Mosimann, 1995; Kneifel and Moisseev, 2020). Thus, in this case, the 1-km thick supercooled liquid stratus layer provides sufficient conditions for snowflakes to rime heavily enough to be classified as graupel. Figure 8d shows the peakTree hydrometeor classification. Rimed particles with more than 1.5 m s<sup>-1</sup> velocity difference to the slowest falling peak are defined as graupel. It is clearly visible how strong riming occurs during seeding before 9 UTC while after 9 UTC and without seeding only the lowest height range shows signal of graupel falling out of the cloud. This compares well with VISSS results found in Fig. 10. At 3-4 km height, there are some points classified as liquid only. These points correspond to the same region where increased spectral width, probability of cloud droplets, rimed mass fraction and DWR were observed. For more details see the discussion on Fig. 5. The percentage of occurrence of the different categories in the hydrometeor classification (for all valid pixels) were as follows: 86.60 % unclassified, 8.07 % liquid, 4.91 % rimed ice, 0.40 % rimed + liquid, 0.01 % columnar ice, and 0.01 % ice + columns. Compared with the VISSS and 2DVD images of the ice crystals there is a good agreement. Almost all particles that were observed at the surface were rimed ice particles, like rimed aggregates and rimed dendrites. There was no liquid precipitation observed. However, the liquid droplets detected by peakTree were small supercooled droplets that obviously stayed within the cloud or were consumed by the surrounding ice.

Figure 9 shows SLDR RHI scans of Mira35 from 90 to 150° elevation angle and polarizability ratio  $\xi$  as derived with the VDPS method. In Fig. 9a) and b) a case with strong seeding (4:08 UTC) is shown. From 6 km to 3 km height, the polarizability ratio derived using the VDPS method is approximately  $\xi \approx 1.3$  (Fig. 9b). This is associated with high values of SLDR (Fig. 9a) indicating the presence of prolate ice crystals at this altitude. The polarizability ratio  $\xi$  decreases from 3 km to 2 km height reaching a value of approximately  $\xi \approx 1$  at around 2.4 km height. This indicates a gradual transformation of prolate ice crystals into isometric ice crystals (either spherical or low-density). The shape transition occurs gradually and thus hints at an aggregation process, resulting in low-density aggregates, characterized by isometric particles. The temperature range from -10 to -15°C supports the interpretation of an aggregation event. However, it cannot be excluded that also a supercooled liquid layer contributed to the transition into more spherical particle shapes via the riming process. Increased probabilities for this process are highlighted in Figs. 5c) and d). Below this layer, there is a population of columnar-shaped crystals, characterized by a polarizability ratio  $\xi \approx 1.3$ . It is separated from the isometric particles above by a shear zone as visible in Fig. 3c. Below 1.3 km height, the polarizability ratio reaches progressively a value of  $\xi \approx 1$ , indicating that particles convert into an isometric shape. This observation is correlated with a high availability of supercooled liquid droplets shown in Fig. 5c and d, leading to the conclusion that a riming event is occurring, producing spherical and dense graupel. This is also in line with enhanced MDV (not shown). The temperature ranged from -3 to  $-10^{\circ}$ C, supporting the interpretation of a riming event. In agreement, the surface observations of ice crystals with VISSS (see Fig. 9c) show a large variety of ice crystal shapes. The observed heavily rimed dendrites as well as needles and multi-modal particle sizes are a result of many different processes mentioned above within the cloud system. Formation and aggregation of dendrites and needles is possible in different regions of the cloud and riming happens in the supercooled liquid layer provided by the Bise cloud.

In case of weak seeding (Figs. 9d and 9e) a very similar feature is visible. Pristine columnar ice crystals formed in the upper part of the cloud at 7 km height and aggregation takes place during the passage of the hydrometeors from cloud top toward the ground. The temperature ranges between  $-30^{\circ}$ C and  $-25^{\circ}$ C at 5.6 to 4.8 km altitude and allows the formation of plate-like ice crystals which dominate the signal and lead to a polarizability ratio  $\xi 

**Figure 9.** SLDR RHI scan of SLDR MBR5 (a, d, g) and corresponding profiles of polarizability ratio  $\xi$  (b, e, h) at 4:08 UTC (strong seeding, a, b), 4:38 UTC (weak seeding, d, e) and 10:38 UTC on 8 Jan 2024 (no seeding, g, h). The black dashed lines indicate the borders between isometric and low density ice particles (with the red dashed line in the center) to oblate and prolate particles. The calculation of errorbars can be found in Teisseire et al. (2025) and Teisseire et al. (2024). The surface observations of the ice crystal shapes with VISSS (in c, f, i) for each 30 s time interval during the radar scan period.

In the case without seeding (g and h), again columnar ice crystals formed within the upper cloud. In Figure 9h, the polarizability ratio decreases between  $5.5\,\mathrm{km}$  and  $3.5\,\mathrm{km}$  height indicating that the particle shapes are transforming toward more isometric ones. The characteristics of the supercooled low cloud layer are different than for cases with seeding. The polarizability ratio is  $\xi\approx 1$  at cloud top and around  $\xi=0.6$  at  $0.8\,\mathrm{km}$  height, indicating that plate-like ice crystals are forming. Below  $0.8\,\mathrm{km}$  height, the polarizability ratio  $\xi$  increases from  $\xi=0.6$  to  $\xi=1$ , reflecting a transformation of ice crystal shape from plate-like to isometric. With the presence of supercooled liquid droplets in the cloud, this transformation in ice crystal shape can be attributed to a riming event. Indeed, measurements with the 2DVD and VISSS as well as documentation entries in an educated-eye protocol support the statement that rimed dendrites are the precipitation type produced by the cloud. Also VISSS measurements in Fig. 9i) indicate that the heavily rimed ice crystals originate from dendrite-shaped ice crystals. The precipitation is very weak, so that only a negligible precipitation amount is measured by the Parsivel<sup>2</sup> disdrometer but nevertheless a few ice crystals are detected by 2DVD and VISSS.

570

# 5.3 In situ: riming dynamics during seeder-feeder interaction

585

590

**Figure 10.** Results of the riming retrieval from Kneifel and Moisseev (2020) and Maherndl et al. (2023, 2024). The sum of rimed mass per 1-minute interval  $m_{\text{rime}}^{\text{sum}}$  (riming rate) is shown in blue and the fraction of rimed mass FR is shown in red (see Sect. 3.12). The case study is from 8 Jan 2024, 0–16 UTC, based on VISSS data.

In the previous parts of this study, it was shown how the evolution of microphysical properties of ice crystals on their way through the cloud can be put into context with ice crystals that are observed at ground with the 2DVD and VISSS. So far, the focus was more on the remote sensing of ice crystals. However, in the following, a more detailed view on particles observed at ground will be shown.

Figure 10 presents results of the VISSS-based riming retrieval from Maherndl et al. (2023, 2024). During periods of strong seeding (mostly between 0 and 3 UTC and between 6 and 9 UTC), the sum of rimed mass rate reaches approximately 10 g min<sup>-1</sup>, with the fraction of rimed mass of around 90%. Later, between 9 and 16 UTC, when there is no interaction between the two cloud systems, the fraction of rimed mass increased to about 95%, while the sum of rimed mass rate decreases to approximately 1 g min<sup>-1</sup>. This higher fraction of rimed mass but lower overall rimed mass rate suggests that the particles were fewer in number and smaller in size but more heavily rimed. We see strong riming signatures between 0 and 9 UTC with particles seeding and growing while falling through the Bise cloud (pronounced signatures in Fig. 8b), and after 9 UTC, there is only a rimed peak in the lowest range gates.

During the morning hours, the ice crystals are larger in size and number, though slightly less rimed compared to the afternoon. These findings are consistent with VISSS and 2DVD surface observations shown in Fig. 6.

# 595 6 Discussion

600

# 6.1 Quantifying precipitation enhancement from seeder-feeder interactions using LWP analysis

Figure 11. LWP and total precipitation on 8 Jan 2024, 0–16 UTC. a) LWP from HATPRO is shown. b) Total precipitation from Parsivel<sup>2</sup> disdrometer is shown.

The presented analysis of the 8-Jan 2024 case study clearly demonstrates that the seeder-feeder interaction can lead to an enhancement of ground-level precipitation in Eriswil (see for example Fig. 6). Figure 10 shows a significant percentage of rimed mass, indicating that a lot of ice mass originated from liquid water. However, the contribution of the seeder-feeder interaction to the overall precipitation enhancement remains uncertain. Figure 11a shows the LWP from HATPRO. Stable background LWP values can be found from 3–5 UTC and from 10–16 UTC. These periods are coinciding with times where no or only weak seeding occurs. Between 0 and 3 UTC and between 6 and 8 UTC the LWP is lower. Exactly at these times it is obvious from Fig. 11 that precipitation occurs. The available LWP might have rimed the ice crystals. As a result the LWP might be reduced.

Overall, it is obvious that during all times the LWP was decreased, the precipitation was increased and vice versa. There is quite a strong contrast not only between seeding and no seeding but also between weak seeding and strong seeding. While the reflectivity in the feeder cloud is very similar for strong seeding and weak seeding (as discussed in Fig. 4) the precipitation shows much lower values for weak seeding than for strong seeding. It seems that a specific amount of seeding mass must fall through the feeder cloud in order to significantly get rimed. Nevertheless, it is not easily possible to calculate an exact contribution of the seeder and feeder cloud to the overall precipitation. Under the assumption that the cloud would regenerate within an hour, the contribution of the feeder cloud to the overall precipitation was assumed to be in the order of 20–40% (not shown), however, with high uncertainty due to the unknown regeneration rate of the LWP reservoir of the feeder cloud.

# 6.2 Precipitation during seeder-feeder events in operational weather models

605

610

620

Figure 12a compares observed and ICON-D2 simulated precipitation rates for the grid point closest to Eriswil and for the nearby locations. Strong natural seeding events occur between 0 and 3 UTC and 5 and 8 UTC, during which all model runs underestimate the observed precipitation. No significant seeding is observed between 3 and 5 UTC and after 9 UTC, and consequently, little to no precipitation is recorded. However, the ICON-D2 model significantly overestimates precipitation during non-seeding periods, simulating nearly 0.2 mm h<sup>-1</sup> of continuous precipitation, caused by simulated ice formation within the Bise cloud. It must be noted that this phenomenon is not only limited to Eriswil. The surrounding places (Huttwil, Napf, Affoltern, and Egolzwil, approximately 15 km around Eriswil) also show significantly higher precipitation during the seeder event and significantly lower precipitation in case of no seeding. This even takes place at Napf which is 1408 m above sea level and therefore within the Bise cloud. This is a strong indication that the model systematically struggles to reproduce precipitation patterns in the vicinity of the Bise cloud in this case study.

The ICON-D2 operational forecast model uses a 1-moment scheme, which only accounts for a simplified ice-nucleating process based on temperature. The differences between forecasted and observed precipitation suggests that fewer INPs may have been present over Eriswil and thus less efficient primary ice formation should be assumed in the simulations.

Figure 12b compares modeled (ICON-D2 mixing ratio of graupel divided by the sum of mixing ratio of graupel, rain and snow) vs radar-retrieval based rimed mass fraction. It is clearly visible that the rimed mass fraction, which is a measure for the presence of graupel, decreases in the observations if no seeding took place anymore. However, in the model the percentage of graupel increases significantly if no seeding took place. In the end, an underestimation of graupel during seeding and an overestimation of graupel during the occurrence of only the Bise cloud was found. This could be an explanation for the respective underestimation of precipitation during seeding and overestimation of precipitation without seeding discussed in Fig. 12a.

The underestimation of precipitation during the seeder-feeder event highlights challenges ICON-D2 faces in accurately simulating this complex process in this case study.

**Figure 12.** a) A comparison of precipitation between ICON-D2 model ensemble runs and observations is presented. Time-lagged ensemble runs are shown as gray curves for the closest grid points to Eriswil. The latest ensemble run starts on 8 Jan 2024, 0 UTC, all other model runs started each 6 hours earlier. The black curve represents observations in Eriswil from the disdrometer, while the colored and dashed lines correspond to precipitation measurements from surrounding locations, as provided by the Kachelmannwetter (Kachelmannwetter, 2024) network (see legend for details, see Fig. 1 for map). The case study is from 8 Jan 2024, 0–16 UTC. b) A comparison of observed vs modeled rimed mass fraction. The observed rimed mass fraction are the averaged results as a profile from the results in Fig. 5d between 0 and 7 UTC ("with seeding: obs." in legend) and between 10 and 16 UTC ("without seeding: obs."). The modeled percentage of graupel is the ratio of graupel content divided by the sum of all precipitation constituents for the model run 8 Jan 2024, 0 UTC where "with seeding: ICON" denotes the average from 0 to 7 UTC and "without seeding: ICON" denotes the average from 10 to 16 UTC.

#### 7 Summary and conclusions

This study presents an in-depth analysis of a seeder-feeder cloud system on 8 Jan 2024 in Eriswil, Switzerland. The conditions are ideal for applying state-of-the-art remote-sensing and in-situ retrieval techniques. Fall streak tracking is applied to get an estimate of microphysical changes of ice crystal properties within cloud systems. Liquid water and riming retrievals show that the Bise cloud contains an extensive reservoir of liquid water that feeds ice crystals that are falling from the seeder cloud through the feeder cloud. This has implications on the observed precipitation enhancement and thus on the water cycle. With a Doppler peak separation algorithm, it is shown that liquid water and ice coexist within the Bise cloud during seeding. In absence of seeding, only negligible amounts of primary ice form in the predominantly supercooled liquid Bise cloud. The transition from pristine ice crystals to aggregates is shown with an ice crystal shape retrieval and then the transition of these aggregates to rimed particles is confirmed with peakTree. These particles are observed in situ at ground where a strong rimed mass fraction is confirmed. In addition, also in situ, higher ICNC are found during seeding. The interaction of seeder and feeder clouds results in a significant precipitation enhancement. It is also found that precipitation is significantly underestimated by the operational ICON-D2 model runs during the seeder-feeder process in this specific case study. It is speculated that the lack of INP within the low-level Bise cloud lead to increased liquid water availability and to stronger riming and precipitation enhancement via enhancing the condensate reservoir for the seeder-feeder process.

The study has several implications. Based on our observations, we found three main insights: The application of multiple advanced remote-sensing methods such as fall streak tracking, Doppler peak separation, and ice shape retrieval shows a consistent view on the case study which highlights the robustness of the methods. This sets an important basis for future studies on cloud processes using similar approaches. Based on the observations, the interactions of the seeder and feeder cloud layers was obtained in unprecedented detail. It was found that the seeder-feeder interaction significantly enhances precipitation which has an impact on the water cycle. From the anti-correlation between surface precipitation and liquid water path we estimated that 20–40 % of the precipitation stems from the feeder cloud. However, we have to note that the value of 20–40 % is strongly dependent on the assumed reproduction rate of liquid water in the feeder cloud. Future studies should focus more on the quantification of the impact of feeder clouds. In the study, also the scientific understanding of microphysical processes like riming and ice crystal shape evolution are deepened. It was found that the ice crystals increased their velocity from around  $-0.8 \,\mathrm{m}\,\mathrm{s}$  to around -1.6 ms once they interacted with the feeder cloud due to the riming process. In the end, the fraction of rimed mass of the particles was around 90 %. The operational ICON-D2 model suggests that key microphysical and dynamical processes are misrepresented in current weather forecast models. Precipitation was significantly underestimated during seeder-feeder phases and overestimated when no interaction occurred. Future studies should look into more seeder-feeder case studies and find if the underestimated effect is a typical feature in such an event. The conclusion would be that a better representation of supercooled liquid water and mixed-phase processes is necessary in order to improve the weather forecast, particularly in regions affected by persistent low-level super-cooled stratus clouds.

Data availability Radar data (scan and vertically pointing), windsond, VISSS (also available by Maahn and Ettrichrätz, 2025), HATPRO, 2DVD are available at Ohneiser et al. (2025). Cloudnet data (including HATPRO, radar data, disdrometer data) can be downloaded from the Cloudnet website (Seifert and O'Connor, 2025). ICON-D2 data is available upon request. The fall streak tracking algorithm can be found in Ohneiser (2025).

**Author contributions** The measurements with LACROS in the frame PolarCAP were collected by KO, PS, WS, TG, VE, AK, HKL, MH, HG, JH, and AS. The measurements in the frame of the Cloudlab project were done by JH, AM, NO, CF, HZ, FR, RS, and UL. All authors contributed to manuscript drafting. The model data was provided by WS and FS. The VDPS method was applied by AT. The fall streak tracking algorithm was applied by KO. The radar data in terms of spectral width, SLDR, DWR, and Doppler spectra, peakTree was analyzed by KO, strongly supported by PS, MR, and TV. The EDR retrieval and radar riming retrieval was conducted by TV. The VISSS data was provided and analyzed by VE, NM, NP, MM, and KO. The 2DVD data was analyzed by TG and KO. The manuscript was written by KO with support by all co-authors.

**Financial support** Funding for this study was provided by the Deutsche Forschungsgemeinschaft (DFG, German Research Foundation) within the priority program SPP 2115 PROM via project numbers 408027490 (PolarCAP, SPOMC) and 408008112 (PICNICC, CORSIPP), by the European Union's Horizon Europe projects CleanCloud (grant no. 101137639), the European Union's Horizon 2020 research and innovation program (CLOUDLAB, grant agreement no. 101021272), and the DFG project EMPOS (project number 516261703). The LACROS infrastructure received financial support via ACTRIS-D, which is funded by the Federal Ministry of Education and Research of Germany under the funding code 01LK2001A.

*Acknowledgements.* We acknowledge Matthias Bauer from Metek company for the great support with the Mira35 MBR5 and MBR7 cloud radars.

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
