# Peer review of "Impact of seeder-feeder cloud interaction on precipitation formation: a case study based on extensive remote-sensing, in-situ and model data"

_EGUsphere, 2025_

## Referee Comment (RC1)

**Review of:** Impact of seeder-feeder cloud interaction on precipitation formation: a case study based on extensive remote-sensing, in-situ and model data

**Authors:** K. Ohneiser, et. al.

**General Comments:**

In general this is an interesting paper. The observational set up is impressive and this may be among the best-observed seeder-feeder situations not involving aircraft. I applaud the authors for their work and fully recognize the amount of effort that went into collecting and interpreting the data.

Having said this, in the end I was a bit underwhelmed with what we actually learned as part of this study. What is new? What did we not know before? What was quantified? With all of these excellent observational platforms, what new perspectives led to new insights? In addition, the model evaluation portion of the work was very underwhelming and resulted in over-generalization of recommendations in the results section.

In the end there is nothing technically wrong with the study and could be published nearly "as is". However, its impact seems likely to be limited without more detail on what the novel findings are as a result of the work put into the collection of this dataset.

**Specific Comments:**

Below I include some specific comments on what is discussed above in the general comments section. Hopefully the authors can consider these comments and assess whether they are able to improve the manuscript as requested based on these comments.

- Line 114: What quantities from the ceilometer are going into VOODOO? Some more detail would be helpful.
- Line 122: Cloudnet ""categorize", or "category" files? Categorize doesn't make sense in context, unless it is the specific name of the files.
- Line 130: Consider defining what is meant by "lowest troposphere".
- Line 152: How do you know that the model is "good"? What does "good" mean? Can you quantify this?
- Section 3.4: Are there any particular considerations for use of HYSPLIT in areas of complex terrain? Are there publications that highlight the performance of this model in such regions? If so, it would be worth highlighting here.

- Section 3.6: It would be useful to know how the use of one algorithm versus another is prioritized in this study. Can you provide additional details?
- Lines 229-230: "Likely presence of only small particles". To what extent has this been evaluated?
- Paragraph starting on line 352: Is the aerosol stuff relevant for the rest of the study? Other than a short mention of INPs, I didn't see any real mention of aerosols. Recommend removing this, along with the associated figure, given that it's not referred to later in the document and it's based on model output.
- Figure 4: The purple and orange curves for the streak tracking don't look as I would anticipate. Are they correct? They don't seem to follow maximum reflectivity, as described earlier in the manuscript.
- Lines 410-411: I do see a slight enhancement of EDR...
- Paragraph starting at line 435: This is an important "check", but I feel like it is over-described. Can this PAMTRA section be condensed to 1-3 sentences?
- Line 467: How are we sure that one of these peaks is not locally-generated ice crystals?
- Figure 7d: There are several "liquid droplets only" points identified around 3-4 km. Yet there is no real discussion of these. Seems noteworthy, given their colocation with some of the other features that are discussed?
- Line 585: "reduced LWP must be the result from being used for riming the ice crystals". First, it's not clear to me that this *has* to be the result of riming only. That's certainly a plausible explanation, but "must be" is very strong language. Second, this sentence structure can be significantly improved (e.g., "The reduced LWP must result from riming of ice crystals.").
- Line 587: What leads to this assumption? Is this based on some calculation? Or just a number pulled out of thin air?
- Section 6.2: I found this section to be severely lacking. Sure, it's nice to *somehow* tie the observational study back to modeling. However, to do this right you would need to look at a variety of different things, including microphysical tendencies, model thermodynamic state, etc. Here this comparison feels like an afterthought that was more included because it was deemed necessary, rather than an actual insightful evaluation of the model and the processes that are (and are not) represented in it. I would recommend removing this alltogether since the paper is already quite long.
- Line 626-627: This conclusion on the modeling is really only for this case, at this location. Is the model even equipped to handle the seeder-feeder process, in terms of parameterizations?

- Line 634-637: This language on model shortcomings is an over generalization. Only one model was evaluated, and only for one case.

**Technical Corrections:**

This section includes corrections and recommendations of a technical nature.

Line 15: "big" seems like an odd word here.... It is used twice. What is "big remote sensing"?

Line 112 and all of manuscript: At times, the word "the" is used excessively. I recommend the authors read through the manuscript again and read sections that include "the" without the word "the" and see if it still makes sense. If so, take it out!

Line 196: Recommend: "enables detection of liquid layers in a cloud."

Line 236: Recommend: "Within updrafts, liquid droplet formation can be enhanced."

Line 317: "Ice ICNC" – use of "ice" is redundant.

Line 426: "probability of riming are visible". Should be either "probabilities" or "is visible".

Line 444: This sentence structure needs some work and doesn't translate correctly in English.

Line 600: "This even takes place..."

---

## Referee Comment (RC2)

The manuscript presented a case study combining remote sensing and in situ data to investigate the potential enhancement of precipitation associated with seeder-feeder mechanism and further compare observations against the model. It is a study with lots of potential. If well-established, it can be applied to more extensive datasets to provide insightful constraints for models. However, there are major issues to be addressed at this point.

You presented multiple seeding profiles during 3-5 UTC in Fig. 4, 6 and 8, but the precipitation rates are very low for this period in Fig. 11. You stated that no seeding occurs between 3 to 5 UTC in your discussion of Fig. 11. These are conflicting. The observations and conclusions are not reconciled.

It is also puzzling that you have 11 profiles selected and shown in Fig. 4, while you picked completely different profiles while you discuss doppler spectrum and polarization ratio etc. in Fig. 6 and 8. A synergetic view means all the coincident observations and available retrievals (reflectivity profile, DWR, retrieved rimed mass, doppler spectrum, etc.) are analyzed and discussed for the identical profiles.

The structure of the paper would be more reasonable, if you presented figures summarizing the entire case (e.g., Fig. 5) for both remote sensing and in situ measurements and then zoom in on representative strong, weak and no seeding vertical profiles and dive into further details.

In addition, the performance of fall streak tracking appears questionable. Related retrievals following the trajectories should be reevaluated. It could have a significant impact on your results. See more in the detailed comments.

The contrast between with and without seeding is evident, but not for the comparison between your strong and weak seeding cases. I expect to see a more in-depth discussion after careful reanalysis of data.

Detailed comments:

Add basic technical details for radars, such as range resolution, minimum detectable sensitivity. Size ranges that 2DVD and VISSS can detect should be provided.

L59: Please add reference for ICON-D2.

L99: reference for peakTree

L122: categorize (verb) → categorized. The same applies to the rest of the manuscript.

L125-126: what is the purpose to obtain PSD and ICNC from both 2DVD and VISSS? Is the information from the 2 instruments complementary or it is for quality control? Also, there is no need to define PSD and ICNC twice.

L153: how large are the uncertainties?

L171: please provide more description in term of how the data is harmonized.

L216: comma should be removed.

L234-235 and L239-240: redundant.

L247-248: How do you define "liquid droplets"? I consider drizzle and rain are part of them.

L254: cloud, drizzle and rain are all liquid water droplets. Raindrops do not fall slowly.

L270: add a table listing the criteria for each classification and providing the percentage of each group – "ice", "ice and liquid", "unclassified", etc. Evaluate how well the classification technique works for your dataset.

L296-297: why not put M in the far left of Eq.2, if that is what to derive?

L304: what is VISSS resolution?

L305: $M$ for individual particles and population should have different symbols.

Eq. 3: is FR for one particle or a population?

L317: VISSS data is not mentioned in section 3.14.

Fig. 4: a panel should be added under panel b to show the time series of liquid water path with error bars indicating retrieval uncertainties.

In Fig. 4b, the first 3 profiles from the left (between 2 and 5 UTC) appear to be under the impact of shear between 1.75 and 2 km by eyeballing the structure of reflectivity. The trajectories are likely slanted. I doubt the particles are falling straight down as indicated. Despite the uncertainties of fall streak tracking, I am not convinced that it works as expected here. I suggest that authors experiment with different time windows when looking for local maximum reflectivity.

L409: what does white color indicate in fig. 5f? It is not part of your colorbar. The high spectral width between 2 and 3 km in fig. 5a mostly collocate with the white colors in fig. 5f. If white indicates missing value, you cannot conclude the high spectral width is not related to turbulence.

L410: remove e.g. in this line.

L459: you state it is a "strong seeding event" here, but your description in Fig. 6 is a "a case with weak seeding".

L460: why not present a case where fall streaks are available and add that profile in Fig. 4?

Fig. 8 caption: the panel letters for weak (c, d) and no seeding (e, f) events are not correct.

L574: are total ICNC from 2 instruments comparable if very small particles below 0.5 mm are excluded?

L583: why is there no seeding between 3 to 5 UTC? You have multiple seeding profiles during this period presented in Fig. 4, 6 and 8.

---

## Author Comment (AC1)

**Reply letter to Reviewer 1.**

**Review of:** Impact of seeder-feeder cloud interaction on precipitation formation: a case study based on extensive remote-sensing, in-situ and model data
**Authors:** K. Ohneiser, et. al.
**General Comments:**
In general this is an interesting paper. The observational set up is impressive and this may be among the best-observed seeder-feeder situations not involving aircraft. I applaud the authors for their work and fully recognize the amount of effort that went into collecting and interpreting the data.
Having said this, in the end I was a bit underwhelmed with what we actually learned as part of this study. What is new? What did we not know before? What was quantified? With all of these excellent observational platforms, what new perspectives led to new insights? In addition, the model evaluation portion of the work was very underwhelming and resulted in over-generalization of recommendations in the results section.
In the end there is nothing technically wrong with the study and could be published nearly "as is". However, its impact seems likely to be limited without more detail on what the novel findings are as a result of the work put into the collection of this dataset.

We thank reviewer #1 very much for providing the review and for the thoughtful comments to the manuscript. (Side note: The line numbers that are being referred to in the responses are according to the revised version without tracked-changes.) The feedback provided us a guideline to improve our conclusions accordingly. We checked our concluding section and found that the text might have been a bit too general, so we went through that part again and added some concrete examples from the manuscript. Now, in our concluding section we have the following paragraph: "The study has several implications. Based on our observations we found three main insights: The application of multiple advanced remote-sensing methods such as fall streak tracking, Doppler peak separation, and ice shape retrieval shows a consistent view on the case study which highlights the robustness of the methods. This sets an important basis for future studies on cloud processes using similar approaches. Based on the observations, the interactions of the seeder and feeder cloud layers was obtained in unprecedented detail. It was found that the seeder-feeder interaction significantly enhances precipitation which has an impact on the water cycle. From the anti-correlation between surface precipitation and liquid water path we estimated that 20-40% of the precipitation stems from the feeder cloud. However, we have to note that the value of 20-40% is strongly dependent on the assumed reproduction rate of liquid water in the feeder cloud. Future studies should focus more on the quantification of the impact of feeder clouds. In the study, also the scientific understanding of microphysical processes like riming and ice crystal shape evolution are deepened. It was found that the ice crystals increased their velocity from around -0.8m s-1 to around -1.6m s-1 once they interacted with the feeder cloud due to the riming process. In the end, the fraction of rimed mass of the particles was around 90%.

The evaluation of output from the operational ICON-D2 model suggests that key microphysical and dynamical processes are misrepresented in current weather forecast models. Precipitation was significantly underestimated during seeder-feeder phases and

overestimated when no interaction occurred. It shows that a better representation of supercooled liquid water and mixed-phase processes is necessary in order to improve the weather forecast, particularly in regions affected by persistent low-level super-cooled stratus clouds."

Below, we'd like to address the specific comments of reviewer #1:

**Specific Comments:**
Below I include some specific comments on what is discussed above in the general comments section. Hopefully the authors can consider these comments and assess whether they are able to improve the manuscript as requested based on these comments.

- Line 114: What quantities from the ceilometer are going into VOODOO? Some more detail would be helpful.

We read through line 114 and the neighboring lines and found that it is not about VOODOO. We are thus not sure which paragraph/section/lines reviewer #1 addresses. The section 3.7 on VOODOO already contains the following paragraph, which to our opinion serves the request of Reviewer #1: "VOODOO utilizes cloud radar Doppler spectra from a vertically pointing radar, attenuated backscatter coefficient at 1064nm from a ceilometer, liquid water path (LWP) retrieved from a microwave radiometer, and temperature, relative humidity, and pressure from numerical weather forecast data from the European Centre for Medium-Range Weather Forecasts (ECMWF)."

- Line 122: Cloudnet ""categorize", or "category" files? Categorize doesn't make sense in context, unless it is the specific name of the files.

Yes, categorize is the specific name of the files in Cloudnet. It is concretized in the manuscript from line 129.

- Line 130: Consider defining what is meant by "lowest troposphere".

In this case, the windsond measurements were available in the lowest 500m above ground. It is now rewritten in the manuscript from line 138.

- Line 152: How do you know that the model is "good"? What does "good" mean? Can you quantify this?

The wording is changed. CAMS can be understood as a leading model in terms of aerosol distribution worldwide. Of course, regional biases can still occur in the model. We included the study of Amarillo et al., 2024 which deals with bias problems in CAMS.

- Section 3.4: Are there any particular considerations for use of HYSPLIT in areas of complex terrain? Are there publications that highlight the performance of this model in such regions? If so, it would be worth highlighting here.

We use the standard HYSPLIT model. There are studies that show the influence of complex terrain on the accuracy of the trajectories. Hernandez-Ceballos et al., 2014 show that the change from GDAS to WRF-ARW enhances the accuracy of the trajectories. Problems in

complex terrain arise when the meteorological forecast data is spatially averaged. We added the information to line 162 in the manuscript.

- Section 3.6: It would be useful to know how the use of one algorithm versus another is prioritized in this study. Can you provide additional details?

We added to the manuscript from line 186: "The wind-based fall streak approach is prioritized if precise wind data with a high resolution are available or the reflectivity field is weak or noisy. In contrast, the reflectivity-based approach is used if reflectivity features are directly associated with hydrometeor descent, or when there is no high-quality wind data available. Presence of directional wind shear remains the largest source of uncertainties.

- Lines 229-230: "Likely presence of only small particles". To what extent has this been evaluated?

The following sentences have been added to the manuscript from line 239: "The size of particles at cloud top has not been evaluated. However, cloud tops typically consist of smaller ice crystals because the cloud top is prone to new particle formation. Therefore, we assume the particles are small at this range. When ice crystals are small enough they are closer to the Rayleigh scattering regime and the difference in scattering between 35 GHz and 94 GHz is minimal, resulting in a near-zero DWR and thus preferable conditions for calibration of the DWR."

- Paragraph starting on line 352: Is the aerosol stuff relevant for the rest of the study? Other than a short mention of INPs, I didn't see any real mention of aerosols. Recommend removing this, along with the associated figure, given that it's not referred to later in the document and it's based on model output.

We understand the concerns of the reviewer #1. Therefore, we removed Fig. 3d. However, we think it is relevant to mention, at least in a short paragraph, that there is dust available in the seeder cloud but not in the feeder cloud. We shortened the paragraph accordingly. The paragraph starts in line 365.

- Figure 4: The purple and orange curves for the streak tracking don't look as I would anticipate. Are they correct? They don't seem to follow maximum reflectivity, as described earlier in the manuscript.

Sometimes, the visual expectation of fall streaks in scenes and the algorithm do not seem to be equivalent, especially if there is strong wind shear like in this case around the border between the PBL and the free troposphere. However, we put a lot of effort into the fall streak tracking algorithm. The code was independently programmed by two different co-authors following the description in the methods section. This yielded the same results. Due to 3-dimensional effects, limitations remain in fall streak tracking, however, it is still a more suitable approach than just vertical profiles. This is already stated in the manuscript from line 195. The difference in visually expected and calculated fall streak may also be a result of the logarithmic color scale.

- Lines 410-411: I do see a slight enhancement of EDR…

Based on the comment of reviewer #1 we evaluated Fig 5f again and can confirm that there is indeed a slight enhancement. We thus removed the part of the sentence "however, here, Fig. 5f indicates that these were not enhanced."

- Paragraph starting at line 435: This is an important "check", but I feel like it is overdescribed. Can this PAMTRA section be condensed to 1-3 sentences?

Based on the reviewer's suggestion, we shortened the paragraph starting in line 442 significantly.

- Line 467: How are we sure that one of these peaks is not locally-generated ice crystals?

We cannot exclude this, although we think it is not the most likely scenario. We included the statement that it is also a possibility that it could be the result of locally-generated ice crystals.

- Figure 7d: There are several "liquid droplets only" points identified around 3-4 km. Yet there is no real discussion of these. Seems noteworthy, given their colocation with some of the other features that are discussed?

Yes, these points correspond to the same features that were discussed in Fig. 5. We referred now to the discussion of that figure.

- Line 585: "reduced LWP must be the result from being used for riming the ice crystals". First, it's not clear to me that this *has* to be the result of riming only. That's certainly a plausible explanation, but "must be" is very strong language. Second, this sentence structure can be significantly improved (e.g., "The reduced LWP must result from riming of ice crystals.").

We thank the reviewer #1 for the recommendation and improved the sentence by replacing "must" by "might" to relax the statement.

- Line 587: What leads to this assumption? Is this based on some calculation? Or just a number pulled out of thin air?

We used peakTree to separate the small cloud droplets from larger cloud particles like drizzle, rain, or ice crystals. We then calculated the average vertical velocity of the small stratus cloud droplets (between 400-1200m, and 0:00-6:30UTC) because we assume they move with the vertical wind. The result was 0.059m/s. We then calculated for an adiabatic cloud how much liquid water content this would produce per time. The result was 4.2g/m^3/min (for calculation of the water vapour saturation pressure we needed the cloud bottom and cloud base temperature which we got from radiosonde profiles). Dividing the stable background LWP between 10 and 16 UTC of 304.5g/m^2 through the liquid water content that can be produced per minute, we get a regeneration rate of 73 minutes. We think the calculation is not very robust and worth to show in the manuscript, therefore we left it out. However, the regeneration rate of approximately an hour is not arbitrarily chosen

and we think that a quantification of the enhanced precipitation through the feeder cloud is a relevant topic for the manuscript. Therefore, we put it into the manuscript and would like to keep it as a motivating feature for future studies.

- Section 6.2: I found this section to be severely lacking. Sure, it's nice to *somehow* tie the observational study back to modeling. However, to do this right you would need to look at a variety of different things, including microphysical tendencies, model thermodynamic state, etc. Here this comparison feels like an afterthought that was more included because it was deemed necessary, rather than an actual insightful evaluation of the model and the processes that are (and are not) represented in it. I would recommend removing this alltogether since the paper is already quite long.
- Line 626-627: This conclusion on the modeling is really only for this case, at this location. Is the model even equipped to handle the seeder-feeder process, in terms of parameterizations?
- Line 634-637: This language on model shortcomings is an over generalization. Only one model was evaluated, and only for one case.

We thank the reviewer #1 for proposing to shorten the manuscript which is already quite long. We nevertheless think that the model evaluation part is quite a central part of the study. Having presented this at conferences, we also found that the model evaluation part triggered discussion and is of high interest for the community. We agree that some statements appeared too generalized. We thus modified the paragraph 6.2 to focus our discussion stronger on the specific case.

Similarly as above, we included statements that this conclusion is only valid for this specific case study (see line 648).

In the entire document, we wrote now that the over-/underestimation can be found for this specific case study. The statement is no longer generalized.

**Technical Corrections:**
This section includes corrections and recommendations of a technical nature.
Line 15: "big" seems like an odd word here…. It is used twice. What is "big remote sensing"?

We removed the word "big" once.

Line 112 and all of manuscript: At times, the word "the" is used excessively. I recommend the authors read through the manuscript again and read sections that include "the" without the word "the" and see if it still makes sense. If so, take it out!

We went through the manuscript and deleted many "the"s if the context was not changed.

Line 196: Recommend: "enables detection of liquid layers in a cloud."

Okay, it is adjusted.

Line 236: Recommend: "Within updrafts, liquid droplet formation can be enhanced."

Okay, it is changed.

Line 317: "Ice ICNC" – use of "ice" is redundant.

Yes, we removed "ice".

Line 426: "probability of riming are visible". Should be either "probabilities" or "is visible".

Changed to "is visible".

Line 444: This sentence structure needs some work and doesn't translate correctly in English.

Okay, the sentence is reformulated to: The observed increase in DWR could even not be reproduced with PAMTRA when assuming 0.5g m^-3 (corresponding to 400g m^-2).

Line 600: "This even takes place…"

Okay, it is corrected.

---

## Author Comment (AC2)

**Reply letter to Reviewer 2.**

The manuscript presented a case study combining remote sensing and in situ data to investigate the potential enhancement of precipitation associated with seeder-feeder mechanism and further compare observations against the model. It is a study with lots of potential. If well-established, it can be applied to more extensive datasets to provide insightful constraints for models. However, there are major issues to be addressed at this point.
You presented multiple seeding profiles during 3-5 UTC in Fig. 4, 6 and 8, but the precipitation rates are very low for this period in Fig. 11. You stated that no seeding occurs between 3 to 5 UTC in your discussion of Fig. 11. These are conflicting. The observations and conclusions are not reconciled.

We kindly refer to the replies provided to the specific comments. (Side note: The line numbers that are being referred to in the responses are according to the revised version without tracked-changes.)

It is also puzzling that you have 11 profiles selected and shown in Fig. 4, while you picked completely different profiles while you discuss doppler spectrum and polarization ratio etc. in Fig. 6 and 8. A synergetic view means all the coincident observations and available retrievals (reflectivity profile, DWR, retrieved rimed mass, doppler spectrum, etc.) are analyzed and discussed for the identical profiles.

See answer in the specific comments.

The structure of the paper would be more reasonable, if you presented figures summarizing the entire case (e.g., Fig. 5) for both remote sensing and in situ measurements and then zoom in on representative strong, weak and no seeding vertical profiles and dive into further details.

As reviewer #2 suggested, we moved the general in-situ part to the general overview and then go into the details afterwards.

In addition, the performance of fall streak tracking appears questionable. Related retrievals following the trajectories should be reevaluated. It could have a significant impact on your results. See more in the detailed comments.

See answer in the specific comments.

The contrast between with and without seeding is evident, but not for the comparison between your strong and weak seeding cases. I expect to see a more in-depth discussion after careful reanalysis of data.

We thank reviewer #2 for this comment. We added the following paragraph to the discussion of Fig. 11 from line 604: "There is quite a strong contrast not only between seeding and no seeding but also between weak seeding and strong seeding. While the reflectivity in the feeder cloud is very similar for strong seeding and weak seeding (as discussed in Fig. 4) the precipitation shows much lower values for weak seeding than for strong seeding. It seems that a specific amount of seeding mass must fall through the feeder cloud in order to significantly get rimed." For this reason, we think that the differentiation between strong and weak seeding is useful.

Detailed comments:

Add basic technical details for radars, such as range resolution, minimum detectable sensitivity. Size ranges that 2DVD and VISSS can detect should be provided.

In Sect. 3.1 we added the following parameters within the text at the respective places:
"The RPG94 minimum detectable sensitivity is -60 dBz down to 50m altitude and the range resolution is 15-30m and 1-3s which can be adjusted by the user."
"The Mira35 minimum detectable sensitivity goes down to -50 dBz at 5km height and range resolution is 30m and 3s. The radar can detect targets down to 150m above ground, with full sensitivity above 450m."
"The resolution of the 2DVD is 0.17mm which corresponds to the smallest detectable particle size."
"The resolution of the VISSS is 0.06mm which corresponds to the smallest detectable particle size."

L59: Please add reference for ICON-D2.

We added Zängl et al.,2015 and Omanovic et al., 2024.

L99: reference for peakTree

Reference is added: Radenz et al., 2019

L122: categorize (verb) categorized. The same applies to the rest of the manuscript.

Categorize is the specific name of the files in Cloudnet. It is concretized in the manuscript from line 129.

L125-126: what is the purpose to obtain PSD and ICNC from both 2DVD and VISSS? Is the information from the 2 instruments complementary or it is for quality control? Also, there is no need to define PSD and ICNC twice.

Both instruments (or at least their application to snowfall in the case of the 2DVD) are relatively new and retrievals are not heavily tested, so we used it as a chance to see if both methods agree well with each other. Also, the instrument setup and resolution are different. VISSS shows a higher ICNC which is the result of its sensitivity to lower particle sizes. It underlines the importance to be able to measure down to very small particle sizes. The Cloudlab campaign provided a unique chance to set bot measurements into relation.
But true, we removed the redefinition of PSD and ICNC.

L153: how large are the uncertainties?

According to the request by reviewer #1 the section was removed from the manuscript. The corresponding Fig. 3d was left out as well.

L171: please provide more description in term of how the data is harmonized.

We added the following paragraph: "For this, the raw data is harmonized, for example into netCDF formats. Then calibration and quality control are done. The Level-2 cloud products generation is performed, including vertical profiles, cloud classification, and liquid/ice water content. Finally, harmonization steps like bias correction, inter-site instrument matching, and consistency checks are done. All processing steps are traceable and reproducible and the datasets get an individual DOI."

L216: comma should be removed.

Comma is removed now.

L234-235 and L239-240: redundant.

Right, the sentence in L239-240 was removed.

L247-248: How do you define "liquid droplets"? I consider drizzle and rain are part of them.

Thanks to reviewer #2. There is a need to be more precise here. We actually meant "liquid cloud droplets". It is corrected in the manuscript.

L254: cloud, drizzle and rain are all liquid water droplets. Raindrops do not fall slowly.

Yes, same as above; we meant liquid cloud droplets. It is corrected in the manuscript in line 260.

L270: add a table listing the criteria for each classification and providing the percentage of each group – "ice", "ice and liquid", "unclassified", etc. Evaluate how well the classification technique works for your dataset.

We added a Tbl. 2 in Sect. 3.9 listing the thresholds for the different categories. Also, we added the percentage of occurrence of the different categories in the hydrometeor classification in the discussion of the respective figure. There, we also discussed how well the results of the peakTree hydrometeor classification agreed with the in-situ observations.

L296-297: why not put M in the far left of Eq.2, if that is what to derive?

The suggestion of reviewer #2 does make sense and we adjusted the equation in the way that M is on the left side of the equation.

L304: what is VISSS resolution?

We changed the formulation to:" resolution of VISSS which is 0.06mm".

L305: *M* for individual particles and population should have different symbols.

Alright, we included the index "i" for the individual particles.

Eq. 3: is FR for one particle or a population?

The equation can be used for both. Here, in the end, it is used for a particle population.

L317: VISSS data is not mentioned in section 3.14.

We thank reviewer #2 for the remark. We removed VISSS data in the subsection title.

Fig. 4: a panel should be added under panel b to show the time series of liquid water path with error bars indicating retrieval uncertainties.

On the one hand, it a good idea of the reviewer #2 to show the evolution of the LWP early. On the other hand, the LWP should be shown together with the precipitation later in the manuscript. If we would include the LWP into Fig. 4, we would show the same LWP curve twice in the manuscript. We think, it is not appropriate to show a figure twice. As a compromise, in the discussion of Fig. 4, we now refer to Fig. 11a containing the LWP measurements (see line 412).

In Fig. 4b, the first 3 profiles from the left (between 2 and 5 UTC) appear to be under the impact of shear between 1.75 and 2 km by eyeballing the structure of reflectivity. The trajectories are likely slanted. I doubt the particles are falling straight down as indicated. Despite the uncertainties of fall streak tracking, I am not convinced that it works as expected here. I suggest that authors experiment with different time windows when looking for local maximum reflectivity.

Sometimes, the visual expectation of fall streaks in scenes and the algorithm do not seem to be equivalent, especially if there is strong wind shear like in this case around the border between the PBL and the free troposphere. However, we put a lot of effort into the fall streak tracking algorithm. The code was independently programmed by two different co-authors following the description in the methods section. This yielded the same results. Due to 3-dimensional effects, limitations remain in fall streak tracking, however, it is still a more suitable approach than just vertical profiles. This is already stated in the manuscript from line 195. The difference in visually expected and calculated fall streak may also be a result of the logarithmic color scale.

L409: what does white color indicate in fig. 5f? It is not part of your colorbar. The high spectral width between 2 and 3 km in fig. 5a mostly collocate with the white colors in fig. 5f. If white indicates missing value, you cannot conclude the high spectral width is not related to turbulence.

The white color is related to data points where no EDR could be retrieved. A fit is calculated to the power-spectrum. If the slope is close to -5/3 (specifically, in between -2 and -1.33, which is 20% uncertainty according to Borque et al. 2016) then the EDR is calculated. Otherwise, no EDR can be calculated. In the presented scenario, this is the case for some of the points in this layer with enhanced spectral width and, by definition, for the entire background. Therefore, it is the same white color (meaning that no EDR could be retrieved). In any way, Reviewer #2 is right, there might be a slightly enhanced EDR. Also, reviewer #1 commented on this, so we removed the concluding part of the sentence "however, here, Fig. 5f indicates that these were not enhanced."

L410: remove e.g. in this line.

Okay, it is removed.

L459: you state it is a "strong seeding event" here, but your description in Fig. 6 is a "a case with weak seeding".

Reviewer #2 is right! The figure caption was not updated after we included another case for the Doppler spectrum. It is corrected now.

L460: why not present a case where fall streaks are available and add that profile in Fig. 4?

In our revised manuscript we now changed the times of the fall streaks in Fig. 4, so that they are equivalent with the times discussed in Fig. 6 and 8.
Just as a side-note: The analysis in Fig. 8 can anyway not be done along fall streak as this technique requires a scan of the radar. While scanning, tracking along fall streak is not possible. However, we decided to use the same initial times in Fig. 4.

Fig. 8 caption: the panel letters for weak (c, d) and no seeding (e, f) events are not correct.

It is corrected now.

L574: are total ICNC from 2 instruments comparable if very small particles below 0.5 mm are excluded?

Yes, the results of the ICNC from both instruments should be equivalent. However, so far there are no extensive comparison studies that show agreement/disagreement, strengths, and limitations of both instruments, so we chose to show both measurements. Due to the optical setup, VISSS is more sensitive to smaller ice particles than 2DVD. This information is important when dealing with the dataset. It underlines the importance to be able to measure down to very small particle sizes.

L583: why is there no seeding between 3 to 5 UTC? You have multiple seeding profiles during this period presented in Fig. 4, 6 and 8.

Thanks for carefully reading the manuscript. It is correct that it was formulated inconsistently. We changed the sentence in the discussion of Fig. 11 to what we actually meant: "These periods are coinciding with times where no or only weak seeding occurs."

---

## Author Response (AR2)

**Reply letter**

We thank the reviewer #1 and reviewer #2 for taking the time to read through the manuscript again and the reviewer #2 for the feedback on the Sect. 6.2. We acknowledge that general statements on the representation of the seeder-feeder effect in the ICON-model cannot be concluded from a single case study. Therefore, we already removed overgeneralizing sentences. We are willing to further reduce the respective paragraphs. Still, we think that the model comparison is quite important and worth to show. Therefore, we propose to leave out all scientific interpretation of the model comparison and move it to an earlier section. In the uploaded new version of the manuscript, we removed the Fig. 11b (along with its discussion) and we moved the Fig. 11a as the new Fig. 4 to the Sect. 4. We removed all scientific interpretation regarding the model comparison and only describe the figure and we briefly raise the potential precipitation underestimation/overestimation in the outlook to trigger for example a Master's thesis on this topic.

In addition, and off-topic, we changed the online-link of the Master's thesis of Gaudek, 2025 to the recently published paper (in ACPD) by Gaudek et al., 2025 as it is the same contents but a more reasonable citation (see line 328).